# LineageFlow: Flow Matching for High-Fidelity Family-Aware Protein Sequence Generation

Langzhang Liang [1 2 3 4]  Ming Yang [4 5]  Yi Feng [3 6]  Junfan Li [7]  Shirui Pan [4 8]  Yinghui Xu [1]  Tianlei Ying [3 6]  Yizhen Zheng [4 9]  Zenglin Xu [1 2]

## Abstract

Protein sequence generation for engineering requires samples that are biophysically plausible and, when targeting a family/domain, remain recognizable members while exploring within-family diversity. Current discrete generative models typically start from uniform or masked-token noise, which discards strong position-specific constraints induced by evolution and forces the model to reconstruct conserved residues from scratch, leading to weak family control and low plausibility. We propose *LineageFlow*, a Dirichlet flow-matching model that initializes generation from lineage priors derived from ancestral sequence reconstruction, turning generation into structured mutation from an evolved scaffold. Across diverse protein families, LineageFlow achieves family validity close to held-out natural sequences and improves predicted structural confidence over uniform-/mask-initialized baselines while maintaining substantial novelty and diversity. Finally, we introduce *rerouting*, a single intermediate-time mutate–select–amplify intervention that enables objective-guided sampling without per-step predictor guidance and yields further gains in plausibility, including a zero-shot enzyme generation case study. Code is available at https://github.com/Jinx-byebye/LineageFlow.

[1]AI[3] Institute, Fudan University [2]Shanghai Academy of AI for Science [3]Shanghai Innovation Institute [4]Theta Team, Australia [5]School of Information and Communication Engineering, Dalian University of Technology [6]School of Basic Medical Sciences, Fudan University [7]Harbin Institute of Technology, Shenzhen [8]School of Information and Communication Technology, Griffith University [9]Department of Cancer Medicine, Monash University. Correspondence to: Tianlei Ying <tlying@fudan.edu.cn>, Yizhen Zheng <Yizhen.Zheng@monash.edu>, Zenglin Xu <zenglin@gmail.com>.

*Proceedings of the 43rd International Conference on Machine Learning*, Seoul, South Korea. PMLR 306, 2026. Copyright 2026 by the author(s).

## 1. Introduction

Protein engineering requires searching sequence space for proteins that (i) remain members of a target family/domain and (ii) exhibit favorable properties (Arnold, 1998; Stemmer, 1994). Deep generative models—ranging from large protein language models (Rives et al., 2021; Elnaggar et al., 2022; Madani et al., 2023) to discrete diffusion and flow-based approaches (Alamdari et al., 2023; Atkinson et al., 2025; Truong Jr & Bepler, 2023)—offer a promising route. Here we study sequence generation under evolutionary constraints: samples should be biophysically plausible and, when conditioned on a specific family/domain, remain recognizable members while exhibiting nontrivial within-family diversity. These requirements are foundational: evolutionary consistency is closely tied to functional relevance, and plausibility (e.g., foldability) is a basic sanity check for protein-like sequences.

A key yet underappreciated design choice in discrete generative processes is the initialization (prior / corruption distribution). Common choices—uniform simplex noise or masked-token corruption—are family-agnostic (Stark et al., 2024; Alamdari et al., 2023). However, family membership is characterized by site-specific evolutionary structure: many sites are highly conserved to maintain structural integrity and biochemical function, while others vary to allow for functional adaptation and conformational diversity (Echave et al., 2016; Tokuriki & Tawfik, 2009). Generic priors erase this structure, forcing the denoiser to reconstruct nearly every residue—including conserved positions—from heavily corrupted states. This "from-scratch synthesis" burden is most severe **early** in the trajectory.

We propose an alternative perspective: generation within a lineage is more naturally formulated as transport from an ancestral distribution rather than synthesis from generic noise. We introduce Lineage-Prior Flow Matching (LineageFlow), which replaces generic priors with phylogeny-informed ancestral priors computed via *ancestral sequence reconstruction* (ASR). Rather than learning to synthesize a protein from scratch, the model learns to mutate variable positions and preserve conserved scaffolds.

To support targeted design, we further introduce Rerouting, an inference-time steering mechanism. Unlike continuous guidance methods that require gradients at every step, Rerouting applies a *single*, directed-evolution–inspired "mutate–select–amplify" intervention within the flow (Arnold, 1998). This steers the trajectory toward user-specified fitness objectives while empirically preserving family validity under our evaluation. Figure 1 provides a schematic overview.

Our main contributions and findings are:

- **Ancestral priors beat generic noise:** Rather than generating protein sequences from noisy priors, we encode ancestral information in priors, turning generation from "from-scratch" denoising under uniform/mask noise into structured mutation from an evolved scaffold.

- **Manifold-preserving guidance via single-step rerouting:** Instead of injecting predictor guidance at every Euler step, rerouting applies a single intermediate-time mutate–select–amplify intervention that steers samples toward objectives while empirically preserving family validity.

- **Near-natural family validity and improved plausibility:** In large-scale experiments across diverse protein families, LineageFlow achieves family validity close to held-out natural sequences with improved plausibility proxies and strong novelty/diversity; rerouting enables a zero-shot enzyme case study on held-out families.

## 2. Related Work

**Protein sequence generative models.** Large protein language models trained on massive corpora learn rich sequence regularities and can be used for unconditional generation or prompted design (Rives et al., 2021; Elnaggar et al., 2022; Madani et al., 2023). More recently, generative processes such as diffusion have been adapted to protein sequences (Alamdari et al., 2023), and flow matching has been developed for discrete sequence design via probability-simplex transport (Stark et al., 2024). Protein Bayesian flow networks provide another likelihood-based discrete generative framework (Atkinson et al., 2025). These approaches differ in modeling assumptions and sampling procedures, but many default to generic priors/corruptions (uniform or masked noise), which can make accurate family control challenging when evolutionary constraints are strong.

**Family- and domain-conditioned generation.** Conditioning mechanisms for protein generation typically provide family information explicitly. One strategy is to attach metadata/control tokens during training and prompt generation at inference (e.g., family tags) (Madani et al., 2023). Another strategy conditions on evolutionary context directly, for instance by encoding an input MSA or a set of related sequences and generating new sequences in that context (Ram & Bepler, 2022; Truong Jr & Bepler, 2023). However, our experiments show that such signals do not necessarily translate into accurate family recognition. In contrast, LineageFlow achieves high-fidelity family-aware generation by initializing the generative process from a family-specific ancestral prior without feeding family labels or an MSA prompt into the denoiser.

**Objective-guided generation and directed evolution.** Beyond matching natural sequence statistics, many protein design tasks aim to optimize external objectives. In protein generation, such objectives are commonly incorporated either by (i) explicit conditioning (e.g., control tags or supervised labels) (Madani et al., 2023), (ii) post-hoc reranking/filtering with learned property predictors (Yang et al., 2019), or (iii) adapting the generator itself via reinforcement learning or model-based optimization (Lutz et al., 2023; Brookes et al., 2019). These strategies can be effective but may trade off against family fidelity if property optimization pulls samples off-manifold. *Directed evolution* provides a complementary experimental paradigm, iteratively mutating and selecting a population to improve function (Arnold, 1998). LineageFlow's rerouting adapts this idea as an intermediate-time inference operator (*mutate–select–amplify*), enabling objective-guided sampling while retaining the family-conditioned generative trajectory.

## 3. Background: Flow Matching on the Probability Simplex

**Notation.** Let $K \in \mathbb{Z}^+$ be the amino-acid alphabet size and let $S_K = \{\mathbf{x} \in \mathbb{R}^K_{\geq 0} : \mathbf{1}^\top \mathbf{x} = 1\}$ denote the probability simplex. We embed a discrete residue $y \in \{1, \ldots, K\}$ as the corresponding vertex $\mathbf{e}_y \in S_K$. Let $\mathcal{H}$ denote a collection of protein families. For each family $h \in \mathcal{H}$ with aligned length $L_h$, define the relaxed sequence space $\mathcal{X}_h := (S_K)^{L_h}$. An element $\mathbf{X} = (\mathbf{x}^{(1)}, \ldots, \mathbf{x}^{(L_h)}) \in \mathcal{X}_h$ represents per-position categorical distributions, and a discrete aligned sequence $\mathbf{s} = (y^{(1)}, \ldots, y^{(L_h)}) \in \{1, \ldots, K\}^{L_h}$ corresponds to the vertex $\mathbf{X}(\mathbf{s}) = (\mathbf{e}_{y^{(1)}}, \ldots, \mathbf{e}_{y^{(L_h)}}) \in \mathcal{X}_h$.

**Problem setup: Family-aware generation as flow matching on simplex.** Let $p_{\text{data}}^{(h)}$ denote the empirical distribution on $\mathcal{X}_h$, observed through samples $\mathbf{X}_1 \sim p_{\text{data}}^{(h)}$. Our goal is to learn a generator $p_\theta^{(h)}$ whose decoded samples match $p_{\text{data}}^{(h)}$; we do so by introducing a tractable prior $q_0^{(h)}$ and learning a neural time-dependent vector field $\hat{\mathbf{v}}(\mathbf{X}, t; \theta)$ (shared across families) that transports $q_0^{(h)}$ toward $p_{\text{data}}^{(h)}$ via flow matching. We treat MSA gaps as missing data (excluded from alphabet) and mask gapped positions in the

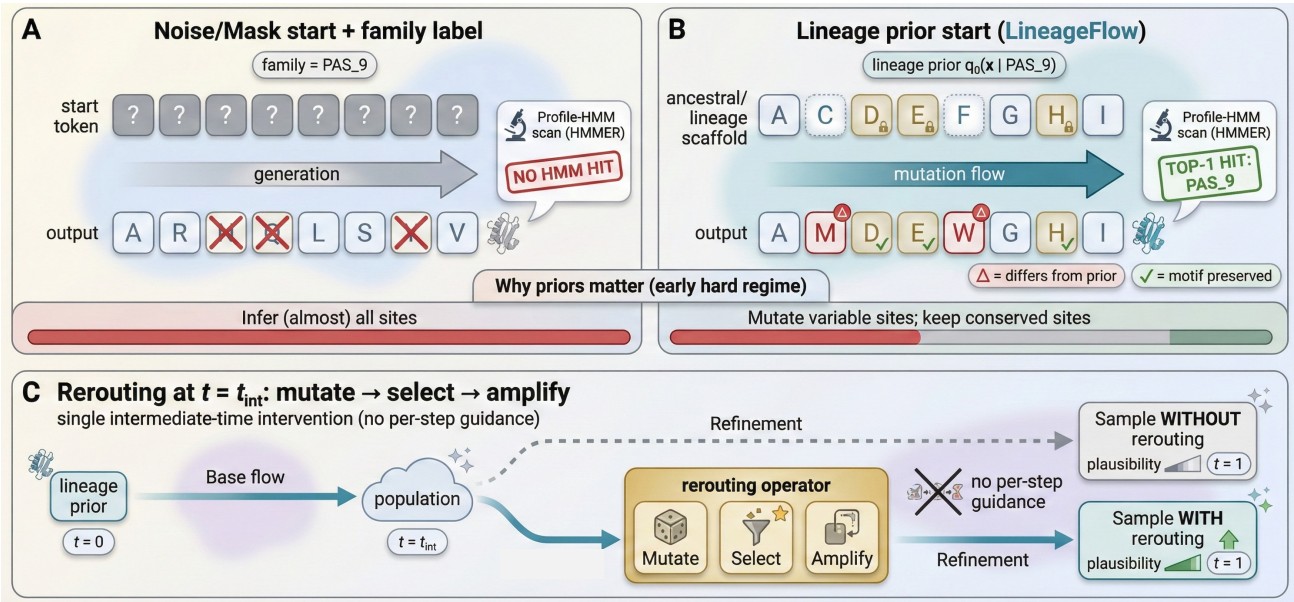

*Figure 1.* **LineageFlow overview. A:** with noise/mask initialization, conditioning on only a family label can still require generating sequences from scratch and can fail to yield a recognizable family-domain sequence. **B:** lineage priors preserve conserved scaffolds, turning generation into structured mutation within a family manifold. **C:** rerouting applies a single intermediate-time mutate–select–amplify intervention for objective-guided sampling without per-step predictor guidance.

probability transport and training loss. To enable variable-length generation, we sample a gap mask from empirical gap rates and keep gaps fixed throughout generation.

**Continuous flow matching.** Fix a family $h$. Let $\mathbf{X}_1 \sim p_{\text{data}}^{(h)}$ be a data sample and $\mathbf{X}_0 \sim q_0^{(h)}$ a prior distribution on $\mathcal{X}_h$. Flow matching (Lipman et al., 2023; Albergo & Vanden-Eijnden, 2023) constructs a generative model via a time-dependent conditional probability path $\{p_t(\mathbf{X} \mid \mathbf{X}_1)\}_{t \in [0,1]}$ satisfying boundary conditions $p_0(\mathbf{X} \mid \mathbf{X}_1) = q_0^{(h)}(\mathbf{X})$ and $p_1(\mathbf{X} \mid \mathbf{X}_1) = \delta(\mathbf{X} - \mathbf{X}_1)$. This path is associated with a conditional vector field $\mathbf{u}_t(\mathbf{X} \mid \mathbf{X}_1)$ solving the continuity equation

$$\partial_t p_t(\mathbf{X} \mid \mathbf{X}_1) + \nabla_{\mathbf{X}} \cdot \big(p_t(\mathbf{X} \mid \mathbf{X}_1)\, \mathbf{u}_t(\mathbf{X} \mid \mathbf{X}_1)\big) = 0.$$

The objective is to learn a neural network $\hat{\mathbf{v}}(\mathbf{X}, t; \theta)$ by minimizing the loss below:

$$\mathcal{L}_{\text{FM}}(\theta) = \mathbb{E}_{\substack{t \sim \mathcal{U}[0,1] \\ \mathbf{X}_1 \sim p_{\text{data}}^{(h)} \\ \mathbf{X} \sim p_t(\cdot \mid \mathbf{X}_1)}} \left[ \big\| \mathbf{u}_t(\mathbf{X} \mid \mathbf{X}_1) - \hat{\mathbf{v}}(\mathbf{X}, t; \theta) \big\|_2^2 \right].$$

**Flow matching on the simplex.** At the level of a single position, the state is a simplex vector $\mathbf{x} \in S_K$, and observed residues correspond to vertices $\mathbf{e}_i \in S_K$. This makes the Dirichlet distribution a natural choice for priors and intermediate distributions on $S_K$.

Dirichlet Flow Matching (DFM) (Stark et al., 2024) defines flows directly on the simplex using a Dirichlet base

distribution $q_0(\mathbf{x}) = \text{Dir}(\mathbf{x}; \boldsymbol{\alpha})$, with density $\text{Dir}(\mathbf{x}; \boldsymbol{\alpha}) = \frac{\Gamma(\alpha_0)}{\prod_{i=1}^{K} \Gamma(\alpha_i)} \prod_{i=1}^{K} x_i^{\alpha_i - 1}$ on $S_K$ (where $\alpha_0 = \sum_{i=1}^{K} \alpha_i$). For $\boldsymbol{\alpha} = \mathbf{1}$, $q_0$ is uniform over $S_K$, with $\mathbb{E}_{q_0}[x_i] = 1/K$ for all $i$. DFM specifies a conditional path in which the Dirichlet mass becomes increasingly concentrated around the target vertex over time:

$$p_t(\mathbf{x} \mid \mathbf{x}_1 = \mathbf{e}_i) = \text{Dir}\big(\mathbf{x};\, \boldsymbol{\alpha} + (t_{\max} t)\, \mathbf{e}_i\big),\ t \in [0,1].$$

Here $t \in [0,1]$ is the normalized flow time and $t_{\max} > 0$ is a fixed scale parameter controlling the terminal concentration (equivalently, one may define an unnormalized time $s = t_{\max} t$). For large $t_{\max}$, $p_{t=1}(\cdot \mid \mathbf{e}_i)$ becomes sharply concentrated around $\mathbf{e}_i$; LineageFlow uses $t_{\max} = 6$.

## 4. Method

**Motivation: Phylogeny-Informed Generative Flow.** Protein sequence space is structured by evolution: modern sequences diverge from shared ancestors, forming distinct family-specific manifolds. Standard phylogenetic theory allows us to infer these ancestral distributions from multiple sequence alignments (MSAs) using statistical models (Eddy, 1998; Yang, 2007). In a discrete generative process, this prior controls how much structure denoising must reconstruct: a uniform or mask prior requires de novo synthesis at every site, whereas an ASR-rooted prior concentrates probability on conserved positions and preserves uncertainty only where the family varies. This reframes generation as structured mutation from an evolved scaffold and provides

a natural family-aligned conditioning signal. Building on this idea, we propose Lineage-Prior Flow Matching (LineageFlow), which uses phylogenetically inferred ancestral priors and learns a simplex-valued flow that transports them toward extant sequences. We interpret the normalized flow time $t \in [0, 1]$ as a proxy for progression along this lineage-informed generative trajectory:

- At $t = 0$ (The Root), the prior is an *Ancestral Distribution*: a site-wise posterior over residues at the root of a phylogenetic tree inferred by ancestral sequence reconstruction (ASR) (Yang, 2007).
- As $t$ increases toward 1 (The Leaves), the flow models the mutational process that transforms this ancestral root into diverse extant sequences, i.e., the sequences in the dataset.
- At an intermediate time $t_{\text{int}} \in (0, 1]$, we apply a directed-evolution *rerouting*—*mutate $\rightarrow$ select $\rightarrow$ amplify*—that biases the population toward a user-defined fitness objective, after which we continue the flow to $t = 1$.

### 4.1. Flow Matching with Ancestral Prior

We represent the $l$th position of a protein sequence as a relaxed categorical vector $\mathbf{x}^{(l)} \in S_K$. For a family $h$ with alignment length $L_h$, the state space is the product simplex $\mathcal{X}_h := (S_K)^{L_h}$.

**Ancestral Initialization (The Prior).** We consider protein sequences partitioned into homologous families indexed by $h \in \mathcal{H}$ (e.g., sequence clusters or domain-based families). Given a family $h$ and its sequences, we build an MSA, infer a maximum-likelihood phylogenetic tree and substitution model (e.g., with IQ-TREE (Nguyen et al., 2015)), and then perform marginal ASR at the root to obtain a site-wise posterior over residues (e.g., with PAML (Yang, 2007)). Rather than using raw MSA column frequencies—which are distorted by phylogenetic redundancy and uneven sampling—or initializing from an extant MSA sequence—which anchors generation near training examples and reduces novelty and diversity—we initialize from the ASR root posterior. By integrating information over the inferred tree under an explicit substitution model, it yields a phylogeny-corrected ancestral scaffold while preserving uncertainty at genuinely variable sites. We encode this site-wise root posterior as Dirichlet concentration parameters $\boldsymbol{\alpha}^{(h,l)} \in \mathbb{R}_{>0}^K$ and define the family-specific ancestral prior

$$q_0(\mathbf{x}^{(l)} \mid h) = \text{Dir}(\mathbf{x}^{(l)}; \boldsymbol{\alpha}^{(h,l)}).$$

where $\mathbf{x}^{(l)} \in S_K$ denotes the relaxed categorical state at position $l$. We use $\boldsymbol{\alpha}^{(h)} := (\boldsymbol{\alpha}^{(h,l)})_{l=1}^{L_h}$ to denote the collection of site-wise priors for family $h$. Details of Dirichlet concentration $\boldsymbol{\alpha}^{(h,l)}$ construction are in Appendix D.

Assuming conditional independence across sites given $h$, the family-specific ancestral distribution over a full sequence $\mathbf{X} = (\mathbf{x}^{(1)}, \ldots, \mathbf{x}^{(L_h)}) \in \mathcal{X}_h := (S_K)^{L_h}$ is

$$q_0^{(h)}(\mathbf{X}) = \prod_{l=1}^{L_h} \text{Dir}(\mathbf{x}^{(l)}; \boldsymbol{\alpha}^{(h,l)}). \tag{1}$$

The global prior over heterogeneous families is a mixture on a disjoint union of state spaces,

$$q_0(\mathbf{X}) = \sum_{h \in \mathcal{H}} \pi_h \, q_0^{(h)}(\mathbf{X}), \quad \pi_h > 0, \sum_h \pi_h = 1. \tag{2}$$

**Lineage-Specific Trajectories.** For a fixed family $h$ and site $l \in \{1, \ldots, L_h\}$, and for a target residue $\mathbf{e}_i$, we define a conditional lineage-specific path that interpolates from the ancestral profile toward the observed data endpoint:

$$p_t^{(h,l)}(\mathbf{x} \mid \mathbf{e}_i) = \text{Dir}(\mathbf{x}; \boldsymbol{\alpha}^{(h,l)} + (t_{\max}t)\,\mathbf{e}_i), \ t \in [0, 1]. \tag{3}$$

We model the continuous transport on the simplex via the vector field:

$$\mathbf{u}_t^{(h,l)}(\mathbf{x} \mid \mathbf{e}_i) = c_h^{(l)}(x_i, t)\,(\mathbf{e}_i - \mathbf{x}). \tag{4}$$

To satisfy the continuity equation (conservation of probability mass along the lineage), the scalar transport speed $c_h^{(l)}$ is derived as (see Appendix B):

$$c_h^{(l)}(z, t) = -\frac{t_{\max} \, \partial_a I_z(a_{h,l}(t), b_{h,l}) \, B(a_{h,l}(t), b_{h,l})}{z^{a_{h,l}(t)-1}(1-z)^{b_{h,l}}}, \tag{5}$$

where $z = x_i$ denotes the probability of the target amino acid, $a_{h,l}(t) = \alpha_i^{(h,l)} + t_{\max}t$, and $b_{h,l} = \sum_{j \neq i} \alpha_j^{(h,l)}$. Here, $B(\cdot, \cdot)$ is the Beta function and $I_z(\cdot, \cdot)$ is the regularized incomplete beta function. Intuitively, $c_h^{(l)}$ acts as a family- and destination-specific speed: it depends on the family-specific parameters $\boldsymbol{\alpha}^{(h,l)}$ and the target residue $\mathbf{e}_i$.

**Training Objective.** We learn the drift via a classifier parameterization: we train a neural network $\hat{p}_\theta(\mathbf{X}_1 \mid \mathbf{X}_t, t)$ that outputs, for each site $l$, a categorical distribution over the terminal residue $\mathbf{x}_1^{(l)} \in \{\mathbf{e}_1, \ldots, \mathbf{e}_K\}$, and then reconstruct the drift field via Eq. (7). Implementation details of the denoiser architecture are provided in Appendix A.2. We minimize the sequence-averaged cross-entropy:

$$\mathcal{L}(\theta) = \mathbb{E}_{h, \mathbf{X}_1, t, \mathbf{X}_t}\left[-\frac{1}{|\mathcal{V}(\mathbf{X}_1)|} \sum_{l \in \mathcal{V}(\mathbf{X}_1)} \log \hat{p}_\theta\left(\mathbf{x}_1^{(l)} \mid \mathbf{X}_t, t\right)\right] \tag{6}$$

where $\mathcal{V}(\mathbf{X}_1)$ denotes the set of valid alignment positions. In classifier-based flow matching, the resulting drift field is reconstructed via:

$$\hat{\mathbf{v}}^{(h,l)}(\mathbf{X}, t; \theta) = \sum_{i=1}^{K} \mathbf{u}_t^{(h,l)}(\mathbf{x}^{(l)} \mid \mathbf{e}_i) \, \hat{p}_\theta \left( \mathbf{x}_1^{(l)} = \mathbf{e}_i \mid \mathbf{X}, t \right).$$
(7)

where $\mathbf{x}^{(l)}$ denotes the $l$th site of $\mathbf{X}$. We define the sequence-level drift by concatenating site drifts, $\hat{\mathbf{V}}^{(h)}(\mathbf{X}, t; \theta) := \left( \hat{\mathbf{v}}^{(h,l)}(\mathbf{X}, t; \theta) \right)_{l=1}^{L_h}$. See Appendix A for pseudocode of the training phase (Algorithm 1).

## 4.2. Generation and Rerouting

We split generation into three phases with a rerouting time $t_{\text{int}} \in [0, 1]$:

- **Base flow** ($t \in [0, t_{\text{int}}]$)**:** lineage-conditioned drift under the learned vector field.
- **Rerouting** ($t = t_{\text{int}}$)**:** discrete rounds of *mutate* → *select* → *amplify* that model artificial selection.
- **Refinement** ($t \in [t_{\text{int}}, 1]$)**:** continued base-flow integration from a selected particle to obtain a full sample.

**Base flow** ($t \in [0, t_{\text{int}}]$)**.** Given a trained model, the base flow follows the usual flow-matching recipe. For *unconditional* sampling, we first draw a lineage index $h \sim \pi$, sample an ancestral sequence $\mathbf{X}_0 \sim q_0^{(h)}$ (Eq. (1)), and integrate the lineage-specific ODE $\dot{\mathbf{X}}_t = \hat{\mathbf{V}}^{(h)}(\mathbf{X}_t, t; \theta)$, from $t = 0$ to $t = t_{\text{int}}$. Different families can have different lengths and unrelated MSA columns. The drift $\hat{\mathbf{V}}^{(h)}$ here is constructed from the classifier using Eq. (7) and the *family-specific* fields in Eq. (4). This procedure defines a baseline distribution at $t_{\text{int}}$ as the mixture of lineage-specific pushforwards induced by $h \sim \pi$ (a distribution over families).

**Rerouting** ($t = t_{\text{int}}$)**: *mutate* → *select* → *amplify*.** For the sampled lineage $h$, the base ODE induces a baseline distribution $p_{t_{\text{int}}}$ on $\mathcal{X}_h$ at the rerouting time. To incorporate a user objective, we treat rerouting as *artificial selection* applied at $t_{\text{int}}$ and realized through rounds of *mutate* → *select* → *amplify*, mirroring directed evolution. Given a mutation/proposal kernel $\mathcal{K}$, a fitness/assay score $J(\mathbf{X})$, and selection stringency $\beta \geq 0$, a single mutate–select round targets the exponentially tilted law

$$p_{t_{\text{int}}}^{\text{sel}}(\mathbf{X}) \propto \left( p_{t_{\text{int}}} \mathcal{K} \right)(\mathbf{X}) \, \exp\left( \beta \, J(\mathbf{X}) \right),$$
(8)

where $p_{t_{\text{int}}} \mathcal{K}$ denotes the proposal law after mutation (taking $\mathcal{K}$ to be the identity kernel recovers the usual exponential tilt of $p_{t_{\text{int}}}$). In practice, we approximate repeated selection by maintaining a population of particles at time $t_{\text{int}}$ and iterating: (i) *mutate* via a proposal kernel $\mathcal{K}$ that injects diversity; (ii) *select* by reweighting particles proportional to $\exp(\beta J)$; and (iii) *amplify* by resampling according to

these weights. $p_{t_{\text{int}}}^{\text{sel}}$ maximizes expected fitness with minimal KL change from the proposal law (Proposition B.3). These rounds update the particle population *at* $t_{\text{int}}$; after the final round, we select a particle $\mathbf{X}_{t_{\text{int}}}^\star$ (e.g., the highest-fitness one). See Appendix A.3 for the concrete fitness and mutation implementations.

**Refinement** ($t \in [t_{\text{int}}, 1]$)**.** Starting from the selected particle $\mathbf{X}_{t_{\text{int}}}^\star$, we continue integrating the base ODE $\dot{\mathbf{X}}_t = \hat{\mathbf{V}}^{(h)}(\mathbf{X}_t, t; \theta)$ up to $t = 1$ to obtain a full generated sample. See Appendix A for inference and rerouting pseudocode (Algorithm 2).

# 5. Theoretical Analysis

We provide theoretical justification for two components of LineageFlow. Proofs are given in Appendix B.

**Proposition 5.1** (Lineage-specific Dirichlet transport)**.** *For any family $h$, site $l$, vertex $\mathbf{e}_i$, and $t \in [0, 1]$, the pair $\left( p_t^{(h,l)}(\cdot \mid \mathbf{e}_i), \mathbf{u}_t^{(h,l)}(\cdot \mid \mathbf{e}_i) \right)$ satisfies the continuity equation on the simplex $S_K$,*

$$\partial_t p_t^{(h,l)} + \nabla_{\mathbf{x}} \cdot \left( p_t^{(h,l)} \mathbf{u}_t^{(h,l)} \right) = 0,$$
(9)

*with zero net boundary flux. Thus the analytic lineage-specific field conserves probability mass and keeps trajectories on the simplex.*

**Proposition 5.2** (Rerouting is KL-regularized selection)**.** *Fix a lineage $h$ and let $p$ be the baseline distribution on $\mathcal{X}_h$ at $t_{\text{int}}$. Let $\mathcal{K}$ be the mutation kernel, $p^{\text{mut}} := p\mathcal{K}$ the proposal law, and $J : \mathcal{X}_h \to \mathbb{R}$ a measurable fitness score. For $\beta > 0$, assume $Z_\beta = \mathbb{E}_{\mathbf{X} \sim p^{\text{mut}}}[\exp(\beta J(\mathbf{X}))] < \infty$ and define*

$$q_\beta(\mathbf{X}) = \frac{1}{Z_\beta} \exp(\beta J(\mathbf{X})) \, p^{\text{mut}}(\mathbf{X}).$$
(10)

*Then $q_\beta$ is the unique maximizer, over $q \ll p^{\text{mut}}$, of*

$$\max_q \left\{ \mathbb{E}_{\mathbf{X} \sim q}\left[ J(\mathbf{X}) \right] - \tfrac{1}{\beta} \mathrm{KL}\left( q \,\|\, p^{\text{mut}} \right) \right\}.$$
(11)

*Moreover, the reweight–resample amplification step consistently approximates $q_\beta$ as the population size grows.*

Proposition 5.1 establishes that the lineage-specific analytic field defines a valid simplex flow. Proposition 5.2 formalizes rerouting as principled selection rather than heuristic guidance: mutation proposes local variants, selection performs an exponential tilt toward high-$J$ regions, and amplification gives a particle approximation of the selected law (full statements and proofs in Propositions B.2 and B.3).

# 6. Experiments

In this section, we evaluate LineageFlow on large-scale protein-domain families, with focus on the research ques-

tions in Box 1.

---

**Box 1.  Research questions addressed in Experiments.**

- **RQ1 (family control):** Can LineageFlow generate sequences that are recognized as belonging to the intended family, compared to (i) explicit family-label conditioning and (ii) strong MSA-prompt conditioning? (Sec. 6.1; Table 1).

- **RQ2 (plausibility):** Does replacing generic noise priors (uniform/mask) with lineage priors improve sequence plausibility, including against a large-scale pretrained generator? (Sec. 6.1; Table 1; Fig. 2).

- **RQ3 (rerouting effects):** How does rerouting at $t_{\text{int}}$ change the sampling distribution and improve plausibility while staying within the family-specific trajectory? (Sec. 6.1; Fig. 3).

- **RQ4 (downstream utility):** Can LineageFlow support zero-shot enzyme-like sequence generation? (Sec. 6.2; Fig. 4).

- **RQ5 (why lineage priors help):** Why do lineage priors beat generic noise priors in protein sequence generation? (Sec. 6.3; Fig. 5).

---

**Dataset and split.**   We train and evaluate on Pfam-A RP35 multiple sequence alignments (Mistry et al., 2021; Chen et al., 2011). After preprocessing (Appendix C), we retain 8,886 families and 8,942,518 aligned sequences. We perform a within-family split, holding out 5% of sequences per family for evaluation.

**Training and generation.**   We train a single classifier $\hat{p}_\theta$ shared across all families using Algorithm 1, sampling $t \sim \mathcal{U}[0, 1]$ with time scale $t_{\max} = 6$. We use 4 NVIDIA RTX 4090 GPUs and train for one epoch (about 26 hours) with learning rate $10^{-5}$ and effective batch size 128. We sample a target family $h \sim \pi$, initialize from its ancestral prior $\mathbf{X}_0 \sim q_0^{(h)}$, integrate the base flow to $t = t_{\text{int}}$, apply rerouting at $t_{\text{int}} = 0.5$, refine to $t = 1$. See Appendix A.4 for the detailed sampling protocol and the held-out baseline.

**Evaluation protocol.**   We follow Appendix E. Family validity measures whether generated sequences are recognized as belonging to their intended Pfam family by profile-HMM scanning (HMMER). Foldability reports predicted structural confidence via OmegaFold (Wu et al., 2022) mean pLDDT. Self-consistency scores the generated sequence under an inverse folding model (ESM-IF (Hsu et al., 2022)) given its predicted backbone. Novelty measures nearest-neighbor sequence identity to the training corpus using

MMseqs2 (Steinegger & Söding, 2017); we report novelty among sequences with pLDDT$\geq$ 70. Diversity measures within-set diversity by clustering foldable sequences (pLDDT$\geq$ 70) with MMseqs2 at 80% identity (coverage 0.8) and reporting the number of clusters.

## 6.1. Main results

**Baselines and conditioning.**   We compare against: (i) ASR-PSSM i.i.d. (sampling directly from the ancestral Dirichlet prior, without any learned flow), (ii) DFM (Stark et al., 2024) (uniform-prior flow matching), (iii) EvoDiff (Alamdari et al., 2023) (mask-based diffusion), and released-weight baselines ProtBFN (Atkinson et al., 2025) and PoET (Truong Jr & Bepler, 2023). We also report a LineageFlow ablation *w/o rerouting* that integrates the base flow without the intermediate mutate–select–amplify step. PoET is conditioned by providing an MSA prompt derived from the intended family at inference time. For DFM/EvoDiff, we enable family conditioning by supplying the target family label to the denoiser; LineageFlow instead conditions through the family-specific ancestral prior $q_0^{(h)}$. Full baseline implementation/training details and external-baseline caveats are provided in Appendix A.5. We generate 1024 sequences for each method.

**Main outcome.**   Table 1 shows that initialization largely determines family-conditional generation quality on Pfam. Uniform-/mask-initialized baselines (DFM, EvoDiff) fail to produce family-consistent, plausible sequences ($\text{Acc}_{\text{fam}} = 0$; low pLDDT), even when the denoiser receives explicit family labels. This supports our premise that generic noise priors force the model to recover nearly all residues from heavily corrupted states, whereas an ASR-rooted prior starts from a family-specific scaffold. Consistent with this, the ASR-PSSM i.i.d. generator (ancestral prior only) already achieves high family validity and reasonable foldability, indicating that the ASR prior carries strong family signal. The base-flow ablation *w/o rerouting* does not improve plausibility over sampling from the ancestral prior, but increases novelty and diversity. Adding rerouting yields near-natural family validity (95.3% vs. 96.6% for held-out natural sequences), improves foldability. Notably, LineageFlow attains higher pLDDT than ProtBFN in our evaluation despite ProtBFN being pretrained on an $\sim 8\times$ larger corpus. For novelty, Novelty@0.8 captures non-replicas of training sequences, while the more stringent Novelty@0.6 reflects deeper novelty while maintaining high family validity; among foldable LineageFlow samples (pLDDT$\geq$ 70), 86.2% satisfy NNId $< 0.8$ and 48.9% satisfy NNId $< 0.6$ (Table 1). Finally, the MSA-prompt PoET baseline shows low Pfam family validity under our metric, underscoring that strong family conditioning signals do not necessarily translate into accurate family recognition. See Ap-

| Method | Family validity | | Foldability | Self-cons. | Novelty & Diversity | | | |
|---|---|---|---|---|---|---|---|---|
| | $\text{Acc}_{\text{fam}}$ ↑ | Hit_any ↑ | pLDDT ↑ | scPPL ↓ | NNId ↓ | Novelty@0.8 ↑ | Novelty@0.6 ↑ | Divers. ↑ |
| Pfam (Held-out) | 96.6 | 100.0 | $86.4 \pm 10.2$ | $5.02 \pm 2.92$ | $0.846 \pm 0.169$ | 33.1 | 12.6 | 806 |
| ProtBFN[†] | — | — | $71.9 \pm 15.7$ | $\mathbf{5.91 \pm 2.75}$ | $0.520 \pm 0.132$ | 93.0 | 64.0 | **604** |
| PoET[‡] | 0.0 | 15.4 | $52.0 \pm 13.5$ | $13.76 \pm 1.85$ | — | — | — | 47 |
| ASR-PSSM (iid) | 92.8 | 98.2 | $70.8 \pm 14.7$ | $7.08 \pm 3.18$ | $0.708 \pm 0.158$ | 72.6 | 32.0 | 378 |
| DFM | 0.0 | 21.9 | $46.2 \pm 15.7$ | $12.62 \pm 2.20$ | — | — | — | 90 |
| EvoDiff | 0.0 | 10.2 | $45.4 \pm 13.4$ | $12.60 \pm 2.43$ | — | — | — | 54 |
| w/o rerouting | 93.0 | 96.9 | $69.6 \pm 14.8$ | $7.96 \pm 3.50$ | $\mathbf{0.602 \pm 0.149}$ | **89.6** | **52.0** | 440 |
| LineageFlow | **95.3** | **99.0** | $\mathbf{76.6 \pm 13.9}$ | $6.67 \pm 3.31$ | $0.620 \pm 0.145$ | 86.2 | 48.9 | 587 |

*Table 1.* Quantitative comparison on generated sequences using the protocols in Appendix E. Family-validity reports profile-HMM top-1 accuracy $\text{Acc}_{\text{fam}}$ and Hit_any (fraction of sequences with any HMM hit at $\tau = 10^{-3}$). Foldability is OmegaFold mean pLDDT (mean±std across sequences), self-consistency is ESM-IF perplexity, and novelty is nearest-neighbor identity to the training corpus with Novelty@$\delta$ reporting the fraction with NNId $< \delta$, computed over sequences with pLDDT$\geq 70$. Diversity is the number of MMseqs2 clusters at 80% identity (coverage 0.8) computed on the foldable subset. † ProtBFN is evaluated by sampling from released weights only (training code not released), and is pretrained on UniProtCC (71M sequences, $\sim 8\times$ larger than our Pfam corpus), so it cannot be family-conditioned; we therefore omit family validity and report its foldability, self-consistency, novelty relative to UniProtCC, and diversity for reference only (Appendix C.1). ‡ PoET is evaluated by sampling from released weights only; at inference time we condition it on an MSA prompt from the intended family (Appendix A.5). Because PoET's training corpus is not released (Appendix C.2), we omit novelty relative to its training corpus. DFM and EvoDiff have no MMseqs hits among the foldable subset, so NNId and Novelty@$\delta$ are undefined and shown as —.

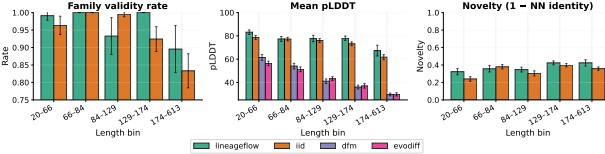

*Figure 2.* **Length-stratified performance.** Pfam unconditional generation metrics as a function of ungapped sequence length (quantile bins). We report mean family validity (profile-HMM top-1), mean pLDDT (OmegaFold), and novelty (1−nearest-neighbor identity; computed on the foldable subset, pLDDT$\geq 70$).

pendix A.5 for potential explanations and additional caveats for PoET/DFM/EvoDiff under this evaluation.

**Length effects.** Figure 2 shows that length is a key driver of generation difficulty: foldability tends to decrease as sequences get longer for all methods. LineageFlow maintains strong family validity across the full length range and exhibits a shallower foldability drop-off than the prior-only baseline, while also producing systematically higher novelty among foldable samples.

**Choosing the rerouting time $t_{\text{int}}$.** We find $t_{\text{int}}$ is a key control knob for directed-evolution-style rerouting: if $t_{\text{int}}$ is too early, the population is highly corrupted and selection acts on unstructured samples; if $t_{\text{int}}$ is too late, simplex states are near one-hot and rerouting has little room to meaningfully change the sequence. Qualitatively, Figure 3 shows that rerouting at $t_{\text{int}}$ can shift the intermediate population toward the region occupied by real family sequences, suggesting that intermediate-time intervention provides a useful

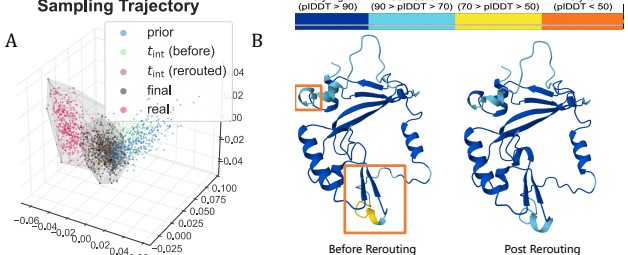

*Figure 3.* **Rerouting shifts the sampling distribution and improves fold confidence (example family PAS_9). (A)** *Sampling trajectory.* We embed generated sequences from four stages of sampling—prior, $t_{\text{int}}$ before rerouting, $t_{\text{int}}$ after rerouting, and final samples—together with real Pfam sequences from the same family, using a simple 3D PCA embedding of $k$-mer features. Rerouting moves the population toward the region occupied by real sequences, illustrating how the intervention changes the distribution at fixed family prior. **(B)** *Structural effect.* Representative predicted 3D structures for sequences sampled at $t_{\text{int}}$ before and after rerouting (colored by pLDDT). Mutated positions are highlighted; rerouting increases predicted confidence (pLDDT).

trade-off between having enough structure for selection to act on, while still leaving room for the population to move. We therefore use intermediate $t_{\text{int}} = 0.5$ in experiments.

Table 2 shows that rerouting increases sampling time relative to sampling without rerouting due to population-based mutate–select–amplify and repeated fitness scoring, while remaining within the same order of magnitude as other Pfam-trained baselines. ProtBFN is substantially slower under its default long-horizon sampling.

| Method | Time (s) | Method | Time (s) |
|---|---|---|---|
| EvoDiff | 352.41 | DFM | 573.63 |
| LF (w/o rerouting) | 759.05 | LF (w/ rerouting) | 1046.83 |
| PoET | 105.32 | ProtBFN | 3724.00 |

*Table 2.* Wall-clock sampling time (seconds) for generating 512 sequences on 4× NVIDIA RTX 4090 GPUs. LF denotes Lineage-Flow.

## 6.2. Zero-shot enzyme generation

To probe downstream utility, we study a *zero-shot* enzyme setting. We hold out three enzyme families—2OG-FEII_OXY, TRP_SYNTA, and RNASE_HII—from denoiser training, but still construct their priors $q_0^{(h)}$ from the corresponding MSAs and phylogenetic trees (Appendix D). At inference, we generate without fine-tuning: $\theta$ is fixed and the only family-specific context is $q_0^{(h)}$. We compare base-flow sampling (no rerouting) to selection-guided rerouting.

We report four family-aware metrics: **(i)** motif agreement, the fraction of conserved profile-HMM match states (threshold $\tau = 0.8$) matching the family consensus; **(ii)** novelty, nearest-neighbor identity (MMseqs2) to the Pfam corpus, where NNId $< 0.6$ indicates deeper novelty among family-recognizable sequences; and **(iii–iv)** solubility and thermostability proxy scores from lightweight ESM-2 (35M) predictors fine-tuned on external datasets (evaluation only): DeepSol and Meltome Atlas.(Khurana et al., 2018; Jarzab et al., 2020) These predictors are imperfect: the solubility predictor achieves strong held-out accuracy, while thermostability is only moderately accurate (about 50% for 5-bin prediction), so these trends should be interpreted cautiously. Across all three families, both base-flow and rerouted samples preserve motifs and remain novel (Figure 4A–B). Rerouting increases motif agreement and shifts solubility and thermostability upward (Figure 4C–D), while keeping NN identities well below typical homology cutoffs. Notably, rerouting optimizes only an unsupervised plausibility objective (ESM2 masked pseudo-likelihood), not the solubility/thermostability predictors. Overall, LineageFlow generates enzyme-like sequences that respect conserved motifs including catalytic residues and match solubility/thermostability proxies of held-out real sequences, without fine-tuning on the target families.

## 6.3. Ancestral priors raise the denoising ceiling in the hard regime

Flow matching relies on a denoiser to infer terminal residues from heavily corrupted intermediate states, so early mistakes propagate through the posterior field used for generation. Figure 5 isolates the effect of initialization by contrasting our family-specific ASR-rooted priors with the uni-

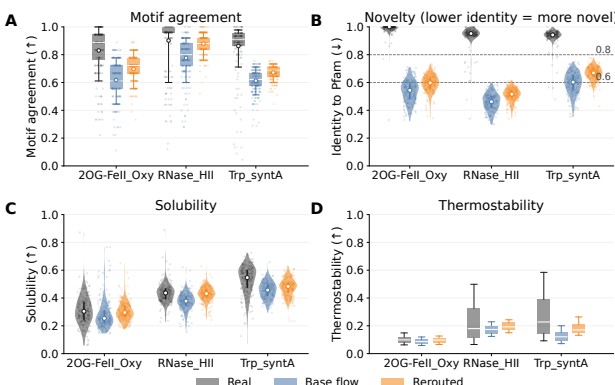

*Figure 4.* **Zero-shot enzyme generation with selection-guided rerouting.** Three held-out enzyme families; we compare held-out real sequences (Real), base-flow sampling (Base flow), and sampling with selection-guided rerouting at $t_{int}$ (Rerouted). **(A)** conservation/motif agreement to the family profile-HMM; **(B)** nearest-neighbor identity to Pfam (lower is more novel; dashed lines mark identity thresholds); **(C)** solubility proxy; **(D)** thermostability proxy.

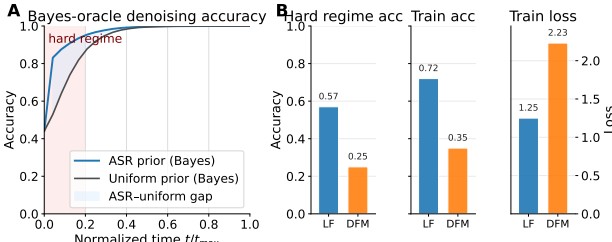

*Figure 5.* **Family-specific ASR priors increase recoverable signal in the hard regime.** **(A)** Bayes-oracle denoising accuracy vs. normalized time $t$ under an ASR prior (LineageFlow) and a uniform prior (DFM). The *pink* region highlights the early-time *hard regime* ($t \leq 0.2$), where $x_t$ is most corrupted. **(B)** Training metrics for LineageFlow (LF) and DFM: hard-regime denoising accuracy (token accuracy in the earliest time bin, $t \leq 0.2$), overall training accuracy, and overall training loss.

form prior used in DFM. In Panel A, the Bayes-optimal decoder is markedly more accurate under ASR priors in the hard regime, indicating a higher recoverable signal in early $x_t$ and thus a higher ceiling for any denoiser (see also Appendix B.1). Panel B summarizes training metrics and shows that LineageFlow attains substantially higher denoising accuracy in the hard regime (token accuracy in the earliest time bin) as well as higher overall training accuracy and lower training loss. Because sampling integrates the learned vector field from $t=0$ to $t=1$, errors made in the hard regime can propagate and dominate final quality even if later-time denoising is strong. Overall, these results support the central mechanism of LineageFlow: *family-specific ancestral priors inject phylogenetic context that reduces early-time ambiguity*, improving the learned posterior field and stabilizing generation.

# 7. Conclusion and Limitations

We presented LineageFlow, a phylogeny-informed generative model that frames protein sequence generation as continuous-time transport from ASR-derived lineage priors on the simplex. Across diverse Pfam families, this lineage-prior initialization yields strong family validity and improved plausibility proxies relative to uniform-/mask-initialized baselines, while maintaining substantial novelty and diversity. Finally, rerouting provides a simple intermediate-time mutate–select–amplify operator for objective-guided sampling without per-step predictor guidance, enabling a zero-shot enzyme generation case study.

LineageFlow relies on high-quality MSAs and phylogenetic inference to construct priors, and generation is tied to family-specific alignment coordinates (including gap masking) rather than explicitly modeling indels and variable-length evolution. Our evaluation uses computational proxies (e.g., OmegaFold pLDDT) and predictor-based property scores without experimental validation, so real-world function remains to be tested. Finally, rerouting introduces additional computational overhead and its effectiveness depends on the fitness function, motivating more robust objectives.

## Impact Statement

This work develops a generative modeling approach for protein sequences that leverages phylogenetic context and a selection-guided rerouting mechanism to steer generation. If used responsibly, such models could accelerate basic research and protein engineering by improving *in silico* exploration of protein families and reducing wet-lab screening burden for benign targets (e.g., enzymes for industrial biocatalysis).

At the same time, protein generation methods can pose dual-use risks: they may lower barriers to designing sequences with harmful properties (e.g., toxins or virulence factors), or be misinterpreted as producing functional proteins without adequate validation. Our results are computational and do not constitute evidence of real-world function or safety; any downstream use should follow established biosafety, biosecurity, and dual-use review practices and include appropriate experimental validation and screening. We encourage restricting applications to clearly beneficial settings and incorporating safeguards such as target selection policies, sequence screening against known hazardous motifs/families.

## Acknowledgements

We would like to thank anonymous reviewers for their constructive feedback.

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

# A. Implementation Details

## A.1. Pseudocode

This subsection provides pseudocode for (i) training the classifier parameterization via supervised flow matching (Algorithm 1) and (ii) sampling with optional rerouting (Algorithm 2). Algorithm 2 follows the three-phase procedure in Sec. 4.2 (base flow → rerouting at $t_{\text{int}}$ → refinement).

---

**Algorithm 1** Training Phase (Supervised Lineage-Prior Flow Matching)

---

1: **Input:**
2:     Dataset $\mathcal{D} = \{(\mathbf{X}_n, h_n)\}_{n=1}^N$, where each $\mathbf{X}_n$ is an aligned sequence of length $L_{h_n}$ with residues in $\{1, \ldots, K\}$ and missing symbols (gaps/unknown)
3:     Family-specific PSSMs $\{\boldsymbol{\alpha}^{(h,l)} \in \mathbb{R}_{>0}^K\}_{h \in \mathcal{H}, \, l=1,\ldots,L_h}$
4:     Neural network classifier $\hat{p}_\theta(\mathbf{X}_1 \mid \mathbf{X}_t, t)$ (sequence in, per-position categorical distribution out)
5:     Time scale $t_{\text{max}}$ (Dirichlet concentration increment)
6:     Learning rate $\eta$
7: **Output:** Trained parameters $\theta$

8: **while** not converged **do**
9:     {1. Sample data and interpolation time}
10:     Sample a training pair $(\mathbf{X}, h) \sim \mathcal{D}$
11:     Let $L_h$ be the aligned length of $\mathbf{X}$
12:     Sample time $t \sim \mathcal{U}[0,1]$
13:     Let $\mathcal{V} \subseteq \{1, \ldots, L_h\}$ be the valid (non-gap/unknown) positions of $\mathbf{X}$
14:     Construct one-hot targets only on valid sites: $\mathbf{x}_1^{(l)} \leftarrow \text{one\_hot}(X^{(l)}) \in \{0,1\}^K$ for $l \in \mathcal{V}$
15:     {2. Sample noisy sequence from family-specific Dirichlet path}
16:     **for** $l = 1$ **to** $L_h$ **do**
17:       If $l \notin \mathcal{V}$, set $\mathbf{x}_t^{(l)} \leftarrow \mathbf{1}/K$ and mark it as missing; **continue**
18:       Compute Dirichlet parameters $\boldsymbol{\alpha}_t^{(h,l)} \leftarrow \boldsymbol{\alpha}^{(h,l)} + (t_{\text{max}}t)\,\mathbf{x}_1^{(l)}$ {Eq. (3)}
19:       Sample noisy site $\mathbf{x}_t^{(l)} \sim \text{Dir}\big(\boldsymbol{\alpha}_t^{(h,l)}\big)$
20:     **end for**
21:     Form noisy sequence $\mathbf{X}_t \leftarrow (\mathbf{x}_t^{(1)}, \ldots, \mathbf{x}_t^{(L_h)})$
22:     {3. Forward pass and loss}
23:     Predict per-position terminal probabilities: $\hat{\mathbf{P}} = \hat{p}_\theta(\mathbf{X}_1 \mid \mathbf{X}_t, t)$ {$\hat{\mathbf{P}} \in [0,1]^{L_h \times K}$}
24:     Compute cross-entropy over valid sites:

$$\mathcal{L}(\theta) = -\frac{1}{|\mathcal{V}|} \sum_{l \in \mathcal{V}} \log \hat{p}_\theta\Big(\mathbf{x}_1^{(l)} \mid \mathbf{X}_t, t\Big) \quad \text{(Eq. (6))}$$

25:     {4. Parameter update}
26:     $\theta \leftarrow \theta - \eta \nabla_\theta \mathcal{L}(\theta)$
27: **end while**

---

---

**Algorithm 2** Inference and Rerouting

---

1: **Input:** Trained network $\hat{p}_\theta$; family priors $\boldsymbol{\alpha}^{(h)}$; time scale $t_{\max}$; rerouting time $t_{\mathrm{int}} \in [0,1]$; steps $N{=}100$ (Euler)
2:     Lineage distribution $\pi$ (or fixed lineage $h^\star$); fitness/assay score $J(\mathbf{X})$; rounds $R$; selection strengths $\{\beta_r\}_{r=1}^R$; population size $M$
3:     Proposal: TOKENDIRICHLETMUTATION$(\mu, \gamma, \rho, \tau_{\mathrm{tok}})$
4: **Output:** Full generated sample $\mathbf{X}_1$

5: **// Phase A: Base Flow to $t_{\mathrm{int}}$**
6: Sample lineage $h \sim \pi$ (or set $h \leftarrow h^\star$)
7: Set $\Delta t \leftarrow 1/N$
8: Set $N_{\mathrm{int}} \leftarrow \lfloor t_{\mathrm{int}}/\Delta t \rfloor$
9: **for** $m = 1$ **to** $M$ **do**
10:     For each site $l = 1, \ldots, L_h$, sample $\mathbf{x}_0^{(m,l)} \sim \mathrm{Dir}(\boldsymbol{\alpha}^{(h,l)})$
11:     Set $\mathbf{X}^{(m)} \leftarrow (\mathbf{x}_0^{(m,1)}, \ldots, \mathbf{x}_0^{(m,L_h)})$
12:     **for** $n = 0$ **to** $N_{\mathrm{int}} - 1$ **do**
13:         Set $t \leftarrow n\,\Delta t$
14:         Predict terminal probabilities: $\hat{\mathbf{P}} = \hat{p}_\theta(\cdot \mid \mathbf{X}^{(m)}, t)$
15:         Compute drift $\hat{\mathbf{V}}^{(h)}(\mathbf{X}^{(m)}, t; \theta)$ via Eq. (7)
16:         Update: $\mathbf{X}^{(m)} \leftarrow \mathbf{X}^{(m)} + \Delta t \cdot \hat{\mathbf{V}}^{(h)}$
17:         Clamp and renormalize each site to enforce $\mathbf{x}^{(l)} \in S_K$; keep gap/missing sites fixed (Appendix A.4).
18:     **end for**
19: **end for**

20: **// Phase B (optional): $R$ rounds of mutate → select → amplify**
21: **for** $r = 1$ **to** $R$ **do**
22:     {Mutate / propose (token-Dirichlet)}
23:     For each particle $m$, sample a mutation mask $S^{(m)} \subseteq \{1, \ldots, L_h\}$ (expected fraction $\mu$; entropy-gated with exponent $\gamma$)
24:     For each $l \in S^{(m)}$, form $q^{(m,l)} \leftarrow \rho\,\hat{p}_\theta(\cdot \mid \mathbf{X}^{(m)}, t_{\mathrm{int}}; \tau_{\mathrm{tok}}) + (1-\rho)\,\boldsymbol{\alpha}^{(h,l)}/\|\boldsymbol{\alpha}^{(h,l)}\|_1$, sample $y^{(m,l)} \sim \mathrm{Cat}(q^{(m,l)})$, and refresh $\mathbf{x}^{(m,l)} \sim \mathrm{Dir}\big(\boldsymbol{\alpha}^{(h,l)} + (t_{\max}t_{\mathrm{int}})\mathbf{e}_{y^{(m,l)}}\big)$
25:     {Select (reweight)}
26:     Compute scores $s_m \leftarrow \beta_r J(\mathbf{X}^{(m)})$ and weights $w_m \leftarrow \exp(s_m - \max_j s_j)$; normalize $w_m \leftarrow w_m / \sum_j w_j$
27:     {Amplify (resample)}
28:     Draw indices $I_1, \ldots, I_M \overset{\text{i.i.d.}}{\sim} \mathrm{Categorical}(w_1, \ldots, w_M)$ and set $\mathbf{X}^{(m)} \leftarrow \mathbf{X}^{(I_m)}$
29: **end for**
30: Select one particle (e.g., $\arg\max_m J(\mathbf{X}^{(m)})$) and denote it $\mathbf{X}_{t_{\mathrm{int}}}^\star$
31: **// Phase C: Refinement to $t = 1$**
32: Set $\mathbf{X} \leftarrow \mathbf{X}_{t_{\mathrm{int}}}^\star$
33: **for** $n = N_{\mathrm{int}}$ **to** $N - 1$ **do**
34:     Set $t \leftarrow n\,\Delta t$
35:     Predict terminal probabilities $\hat{\mathbf{P}} = \hat{p}_\theta(\cdot \mid \mathbf{X}, t)$
36:     Compute drift $\hat{\mathbf{V}}^{(h)}(\mathbf{X}, t; \theta)$ via Eq. (7)
37:     Update: $\mathbf{X} \leftarrow \mathbf{X} + \Delta t \cdot \hat{\mathbf{V}}^{(h)}$
38:     Clamp and renormalize each site to enforce $\mathbf{x}^{(l)} \in S_K$; keep gap/missing sites fixed (Appendix A.4).
39: **end for**
40: **return** $\mathbf{X}$

---

## A.2. Denoiser architecture and simplex embedding

Our classifier $\hat{p}_\theta(\mathbf{X}_1 \mid \mathbf{X}_t, t)$ is implemented as a transformer encoder initialized from the ESM2 checkpoint `facebook/esm2_t33_650M_UR50D`. The model consumes a simplex sequence $\mathbf{X}_t \in (S_K)^{L_h}$ (with $K{=}20$) and outputs per-position logits $\ell^{(l)} \in \mathbb{R}^K$, yielding $\hat{p}_\theta(\mathbf{x}_1^{(l)} = \mathbf{e}_a \mid \mathbf{X}_t, t) = \mathrm{softmax}(\ell^{(l)})_a$.

To embed continuous simplex inputs, we use the expectation under ESM2's input embedding table: letting $\mathbf{w}_a$ denote the

ESM2 input embedding of amino acid $a$, we form $\mathbf{h}_0^{(l)} = \sum_{a=1}^{K} x_a^{(l)} \mathbf{w}_a$ for each position $l$. Gap/unknown positions are represented by setting $\mathbf{x}^{(l)}$ to the uniform vector and passing a boolean `gap_flag`; these positions are excluded from the loss and their drift is masked to zero (Appendix A.4). Time conditioning uses a sinusoidal embedding followed by an MLP, added as a per-token bias.

For DFM/EvoDiff baselines, we optionally add a learned family-ID embedding as a per-sequence bias (broadcast across positions), as described in Appendix A.5.

### A.3. Rerouting implementation

**Fitness function implementation.** For rerouting in Table 1 and Sec. 6.2, we instantiate the fitness objective $J(\mathbf{X})$ as a *soft-masked ESM2* plausibility score computed directly on simplex states (dropping gap positions). Let $\mathbf{X} = (\mathbf{x}^{(1)}, \ldots, \mathbf{x}^{(L)}) \in (S_K)^L$ denote an ungapped simplex sequence of length $L$ (with $K = 20$) and let $\mathcal{M} \subseteq \{1, \ldots, L\}$ be a set of masked residue indices. We form ESM2 inputs using the `[MASK]` embedding at masked positions and the expected amino-acid embedding $\sum_{a=1}^{K} x_a^{(l)} \mathbf{w}_a$ at unmasked positions, where $\mathbf{w}_a$ is the ESM2 input embedding for amino acid $a$. If $\ell^{(l)} \in \mathbb{R}^K$ are the resulting ESM2 logits and $p_{\text{ESM}}^{(l)} = \text{softmax}(\ell^{(l)})$, we score masked positions by the expected log-probability under $\mathbf{X}$,

$$S(\mathbf{X}; \mathcal{M}) \;=\; \frac{1}{|\mathcal{M}|} \sum_{l \in \mathcal{M}} \sum_{a=1}^{K} x_a^{(l)} \log p_{\text{ESM}}^{(l)}(a), \tag{12}$$

i.e., a per-residue mean (normalization enabled). We use the hybrid rerouting score $J(\mathbf{X}_{\text{mut}}) = S_{\text{global}}(\mathbf{X}_{\text{mut}}) + \Delta S(\mathbf{X}_{\text{mut}}, \mathbf{X}_{\text{base}})$, where $S_{\text{global}}$ averages $S(\cdot; \mathcal{M})$ over $G$ random masks with mask fraction $p_{\text{mask}}$, and the local mutation effect is

$$\Delta S \;=\; S(\mathbf{X}_{\text{mut}}; C) \;-\; S(\mathbf{X}_{\text{base}}; C), \qquad C = \left\{ l : \arg\max \mathbf{x}^{(l)} \neq \arg\max \mathbf{x}_{\text{base}}^{(l)} \text{ or } \tfrac{1}{2}\|\mathbf{x}^{(l)} - \mathbf{x}_{\text{base}}^{(l)}\|_1 > \delta \right\}. \tag{13}$$

We use ESM2 `facebook/esm2_t30_150M_UR50D` with $p_{\text{mask}} = 0.15$, $G = 8$, and $\delta = 0.1$.

**Mutation operator implementation.** The mutation/proposal kernel $\mathcal{K}$ in Sec. 4.2 is instantiated as TOKENDIRICHLET-MUTATION$(\mu, \gamma, \rho, \tau_{\text{tok}})$ (Algorithm 2). Given a lineage $h$ and a simplex state $\mathbf{X} \in \mathcal{X}_h$ at $t_{\text{int}}$, we first sample a mutation mask $S \subseteq \{1, \ldots, L_h\}$ over valid (non-gap) positions with expected fraction $\mu$, biased toward high-entropy sites under the prior mean $\boldsymbol{\alpha}^{(h,l)}/\|\boldsymbol{\alpha}^{(h,l)}\|_1$:

$$p_l \;\propto\; \left( \frac{H(\boldsymbol{\alpha}^{(h,l)}/\|\boldsymbol{\alpha}^{(h,l)}\|_1)}{\log K} \right)^{\gamma}, \tag{14}$$

where $H(\cdot)$ is Shannon entropy and the proportionality constant is chosen so that the average mutation probability over valid sites is approximately $\mu$ (with clipping to $[0, 1]$). For each mutated site $l \in S$, we form a token distribution

$$q^{(l)} \;=\; \rho\, \text{softmax}(\ell^{(l)}/\tau_{\text{tok}}) \;+\; (1 - \rho)\, \boldsymbol{\alpha}^{(h,l)}/\|\boldsymbol{\alpha}^{(h,l)}\|_1, \tag{15}$$

where $\ell^{(l)}$ are the denoiser logits at $(\mathbf{X}, t_{\text{int}})$, sample $y^{(l)} \sim \text{Cat}(q^{(l)})$, and refresh the simplex state by

$$\mathbf{x}^{(l)} \sim \text{Dir}\big(\boldsymbol{\alpha}^{(h,l)} + (t_{\max} t_{\text{int}})\mathbf{e}_{y^{(l)}}\big). \tag{16}$$

Non-mutated sites are left unchanged.

**Rerouting hyperparameters.** Unless otherwise noted, rerouting uses $R = 3$ rounds with population size $M = 8$, rerouting time $t_{\text{int}} = 0.5$, and selection strength $\beta_r \equiv 4.0$. We use TOKENDIRICHLETMUTATION with mutation fraction $\mu = 0.25$, entropy-gating exponent $\gamma = 1.0$, posterior mixing $\rho = 0.8$, token temperature $\tau_{\text{tok}} = 1.0$, and token sampling (rather than argmax), together with systematic resampling.

### A.4. Generation protocol and held-out baseline

For Table 1, we generate $N{=}1024$ sequences. For each sequence, we sample a lineage $h \sim \pi$ (Eq. (2)), where we use a smoothed family-frequency distribution $\pi_h \propto n_h^\tau$ with $\tau = 0.5$ and $n_h$ the number of training sequences in family $h$. For computational efficiency, we cap the number of distinct sampled families at 128 while preserving the relative weights under $\pi$. Given $h$, we sample an ancestral initialization $\mathbf{X}_0 \sim q_0^{(h)}$ (Eq. (1)) and generate using Algorithm 2. Unless otherwise

noted, Table 1 uses rerouting with $t_{\text{int}} = 0.5$ and the hyperparameters listed in Appendix A.3. We integrate the ODE with a fixed-step Euler solver using 100 total steps over $t \in [0, 1]$, and decode the final simplex states by per-position argmax (temperature 1). After each Euler step we enforce simplex constraints by clamping and renormalizing $\mathbf{x}^{(l)} \in S_K$, and we keep gap positions fixed as uniform. We compute the scalar speed $c_h^{(l)}(z, t)$ in Eq. (5) using a numerically stable incomplete-beta implementation with $z$ clamped to $[\varepsilon, 1 - \varepsilon]$. We model alignment gaps by sampling a family-specific gap mask from empirical gap statistics estimated on the training split, and keep gap positions fixed throughout base-flow integration and selection; for evaluation, we remove gaps to obtain ungapped amino-acid sequences of variable length.

**Gap mask sampling and handling.** For each family $h$ and alignment column $l$, we estimate the per-column missing rate $m_l^{(h)}$ on the training split (counting – and X as missing; Appendix C). For each generated sequence, we sample a fixed gap mask independently across columns, $\text{gap}^{(l)} \sim \text{Bernoulli}(m_l^{(h)})$, and keep these positions fixed for the full trajectory. Because our simplex state uses $K{=}20$ (no gap token), we represent gap positions by setting $\mathbf{x}^{(l)}$ to the uniform vector in $S_K$ and passing the boolean gap mask as a separate gap_flag input to the denoiser; we also mask the drift so that gap positions have zero update. At evaluation time, we remove gap positions from the decoded sequence. As a reference baseline, we report the same metrics on an equally sized sample of held-out sequences, drawn to match the generated family distribution under $\pi$.

### A.5. Baseline implementations and conditioning

**Pfam-trained baselines.** We include an ASR-PSSM i.i.d. baseline that samples each site independently from the family-specific ancestral prior (with the same gap masking), i.e., no denoiser and no learned flow, to isolate the contribution of the prior alone. DFM (Stark et al., 2024) is implemented as a uniform-prior ablation of our pipeline: it uses the same denoiser backbone and training loop as LineageFlow, but replaces the family-specific ancestral prior with a uniform Dirichlet prior and disables rerouting. EvoDiff (Alamdari et al., 2023) is adapted from the official implementation and uses masked-token corruption. To enable family-conditioned generation for DFM and EvoDiff, we provide the target family label to the denoiser during training and inference; concretely, we embed the family ID and add it as a per-sequence bias to token representations (broadcast across positions). In contrast, LineageFlow does not take explicit family labels and instead conditions via the family-specific prior $q_0^{(h)}$. All Pfam-trained methods are trained on the same Pfam dataset for one epoch; LineageFlow/DFM use ~650M-parameter denoisers and EvoDiff uses a ~500M-parameter denoiser.

**Released-weight baselines.** We additionally include ProtBFN (Atkinson et al., 2025) by directly sampling from released weights; the official release does not include training code, so we cannot retrain it on Pfam or add family conditioning. ProtBFN is pretrained on UniProtCC (71M sequences; Appendix C.1), so it cannot be family-conditioned; we omit family validity and report novelty relative to UniProtCC for reference (Table 1). We also include PoET (Truong Jr & Bepler, 2023) as an *MSA-prompt* baseline by sampling from released weights and conditioning on an MSA prompt derived from the intended family. For each target family $h$, we provide PoET with a prompt formed from the aligned Pfam MSA for $h$ (RP35 filtered) and sample sequences autoregressively conditioned on this prompt. We follow PoET's default prompt construction and budgets: prompt sequences are selected with the neighbor sampler ($\theta = 0.2$) and truncated to at most 128 sequences and 24,576 total prompt tokens per family. We sample with temperature 1 and no top-$k/p$ truncation. Since PoET emits ungapped amino-acid sequences, we set the per-family maximum generation length to the median ungapped length of the family alignment (capped at 1024) to avoid forcing residues into gap-heavy alignment columns. Because PoET's training code and training corpus are not released (Appendix C.2), we omit novelty relative to its training corpus and report family validity, foldability, self-consistency, and within-set diversity for reference.

Note that Pfam-trained methods are generated under the same family sampler $h \sim \pi_h \propto n_h^{0.5}$ (Appendix A.4); released-weight baselines are generated with their released inference pipelines, so their length distributions can differ.

*Remark* A.1 (Why family-aware baselines can fail at Pfam scale). Family validity in our setting is a strict evaluation: each sample must place its intended family as the top profile-HMM match among thousands of families, including shallow families and many close neighbors. Under generic noise priors (DFM/EvoDiff), the denoiser must infer conserved motifs and domain signatures from heavily corrupted states; a global family label provides only coarse information and does not supply site-wise evolutionary context, so generation can drift off family manifolds and yield sequences with no Pfam HMM hit. PoET conditions via an MSA prompt, but this is an *in-context* interface rather than an explicit family prior: generation is not constrained to Pfam's family-specific coordinate system (domain boundaries, gap patterns), and the model still performs from-scratch synthesis of conserved positions. Moreover, PoET is pretrained on UniRef50-derived homology sets rather

than Pfam domain MSAs (Appendix C.2), so prompting with Pfam alignments can constitute a distribution shift. Consistent with these factors, PoET exhibits low Pfam hit coverage in our experiments, suggesting that strong prompts alone do not guarantee accurate family recognition under profile-HMM evaluation at this scale.

# B. Proofs

## B.1. Denoising ceiling monotonicity under conditioning

**Lemma B.1.** *Let $Y \in \{1, \ldots, K\}$ be a discrete label and let $(X, Z)$ be any random variables on a common probability space. Define the Bayes-optimal 0–1 accuracies*

$$A(X) := \mathbb{E}_X \left[ \max_{i \in \{1,\ldots,K\}} \mathbb{P}(Y = i \mid X) \right], \qquad A(X, Z) := \mathbb{E}_{X,Z} \left[ \max_{i \in \{1,\ldots,K\}} \mathbb{P}(Y = i \mid X, Z) \right]. \tag{17}$$

*Then $A(X, Z) \geq A(X)$.*

*Proof.* By the tower property of conditional expectation, for each $i$ we have

$$\mathbb{P}(Y = i \mid X) = \mathbb{E}_Z \big[ \mathbb{P}(Y = i \mid X, Z) \big]. \tag{18}$$

Applying Jensen's inequality conditional on $X$ yields

$$\max_{i \in \{1,\ldots,K\}} \mathbb{P}(Y = i \mid X) = \max_{i \in \{1,\ldots,K\}} \mathbb{E}_Z [\mathbb{P}(Y = i \mid X, Z)] \leq \mathbb{E}_Z \left[ \max_{i \in \{1,\ldots,K\}} \mathbb{P}(Y = i \mid X, Z) \right]. \tag{19}$$

Taking expectation over $X$ gives $A(X) \leq A(X, Z)$, proving the claim. □

In our setting, take $Y$ to be the terminal residue identity at a site, $X = X_t$ the corrupted simplex state at time $t$, and $Z$ the phylogenetic context (lineage and ASR prior parameters). The lemma shows that conditioning on ancestral context cannot decrease the Bayes-optimal denoising accuracy, motivating the hard-regime analysis in Sec. 6.3 (Figure 5).

## B.2. Correctness of lineage-specific transport

**Proposition B.2.** *For any family $h$, site $l \in \{1, \ldots, L_h\}$, vertex $\mathbf{e}_i$, and $t \in [0, 1]$, the pair $\big(p_t^{(h,l)}(\cdot \mid \mathbf{e}_i), \mathbf{u}_t^{(h,l)}(\cdot \mid \mathbf{e}_i)\big)$ satisfies the continuity equation on the simplex $S_K$:*

$$\partial_t p_t^{(h,l)}(\mathbf{x} \mid \mathbf{e}_i) + \nabla_{\mathbf{x}} \cdot \left( p_t^{(h,l)}(\mathbf{x} \mid \mathbf{e}_i) \, \mathbf{u}_t^{(h,l)}(\mathbf{x} \mid \mathbf{e}_i) \right)$$
$$= 0 \quad \text{for all } \mathbf{x} \in S_K. \tag{20}$$

*Moreover, the net boundary flux on $S_K$ vanishes, ensuring that total probability mass is conserved ($\int_{S_K} p_t^{(h,l)} d\mathbf{x} = 1$) throughout the process.*

*Proof.* Fix a family $h$, a site $l \in \{1, \ldots, L_h\}$, and a target vertex $\mathbf{x}_1 = \mathbf{e}_i$. Recall the definitions from the main text:

$$p_t^{(h,l)}(\mathbf{x} \mid \mathbf{e}_i) = \text{Dir}\big(\mathbf{x}; \boldsymbol{\alpha}^{(h,l)} + (t_{\max} t) \, \mathbf{e}_i \big), \tag{21}$$
$$\mathbf{u}_t^{(h,l)}(\mathbf{x} \mid \mathbf{e}_i) = c_h^{(l)}(x_i, t) \, (\mathbf{e}_i - \mathbf{x}). \tag{22}$$

The vector field $\mathbf{u}_t^{(h,l)}$ is defined on the simplex $S_K$. To verify the continuity equation

$$\partial_t p_t^{(h,l)} + \nabla_{\mathbf{x}} \cdot \big( p_t^{(h,l)} \mathbf{u}_t^{(h,l)} \big) = 0, \tag{23}$$

we first exploit the geometric symmetry of the flow. Observe that the vector field is purely radial centered at the target vertex $\mathbf{e}_i$. For any two non-target indices $j, k \neq i$, the equations of motion are $\dot{x}_j = -c_h^{(l)}(x_i, t) x_j$ and $\dot{x}_k = -c_h^{(l)}(x_i, t) x_k$. Consequently, the ratio $x_j / x_k$ is invariant in time:

$$\frac{d}{dt} \left( \frac{x_j}{x_k} \right) = \frac{\dot{x}_j x_k - x_j \dot{x}_k}{x_k^2} = \frac{-c_h^{(l)} x_j x_k - x_j (-c_h^{(l)} x_k)}{x_k^2} = 0. \tag{24}$$

Equivalently, define the normalized non-target proportions $r_j := x_j/(1 - x_i)$ for $j \neq i$. Since $\dot{x}_j = -c_h^{(l)}(x_i, t)x_j$ and $\frac{d}{dt}(1 - x_i) = -\dot{x}_i = -c_h^{(l)}(x_i, t)(1 - x_i)$, we have $\dot{r}_j = 0$ and thus $r = (r_j)_{j \neq i}$ is invariant along the flow. Moreover, the Dirichlet distribution admits the standard Beta–Dirichlet factorization: if $x_i \sim \text{Beta}(a_{h,l}(t), b_{h,l})$ and $r \sim \text{Dir}(\boldsymbol{\alpha}_{-i}^{(h,l)})$ independently, then $\mathbf{x}_{-i} = (1 - x_i)r$ has joint law $\text{Dir}(\boldsymbol{\alpha}^{(h,l)} + (t_{\max}t)\,\mathbf{e}_i)$. Therefore only the Beta marginal of $x_i$ evolves with $t$, while the conditional law of $r$ given $x_i$ is time-invariant. This justifies reducing the continuity equation on $S_K$ to the 1D continuity equation for the marginal density of $x_i$.

Let $f_t(x)$ denote the marginal density of the $i$-th coordinate $x_i$ under the Dirichlet distribution $p_t^{(h,l)}$. The properties of the Dirichlet imply that $x_i$ follows a Beta distribution:

$$f_t(x) = \text{Beta}\big(x; a_{h,l}(t), b_{h,l}\big) = \frac{x^{a_{h,l}(t)-1}(1 - x)^{b_{h,l}-1}}{B\big(a_{h,l}(t), b_{h,l}\big)}, \tag{25}$$

where $a_{h,l}(t) = \alpha_i^{(h,l)} + t_{\max}t$ and $b_{h,l} = \sum_{j \neq i} \alpha_j^{(h,l)}$. We can express the cumulative distribution function (CDF) using the regularized incomplete beta function $I_x(a, b)$:

$$F_t(x) = \int_0^x f_t(z)\,dz = I_x\big(a_{h,l}(t), b_{h,l}\big). \tag{26}$$

The density is the spatial derivative: $f_t(x) = \partial_x I_x\big(a_{h,l}(t), b_{h,l}\big)$.

The scalar velocity of the coordinate $x_i$ induced by the vector field is the projection onto the $i$-th axis:

$$v(x, t) := [\mathbf{u}_t^{(h,l)}(\mathbf{x})]_i = c_h^{(l)}(x, t)(1 - x). \tag{27}$$

We must satisfy the 1D continuity equation:

$$\partial_t f_t(x) + \partial_x\Big(f_t(x)v(x, t)\Big) = 0. \tag{28}$$

First, we compute the time derivative of the density. Since $a_{h,l}(t) = \alpha_i^{(h,l)} + t_{\max}t$ with $da_{h,l}(t)/dt = t_{\max}$:

$$\partial_t f_t(x) = \partial_t\big[\partial_x I_x(a_{h,l}(t), b_{h,l})\big] = t_{\max}\,\partial_a\big[\partial_x I_x(a, b_{h,l})\big]\Big|_{a=a_{h,l}(t)}. \tag{29}$$

By Schwarz's theorem, we interchange the mixed partial derivatives $\partial_a\partial_x = \partial_x\partial_a$:

$$\partial_t f_t(x) = t_{\max}\,\partial_x\Big[\partial_a I_x\big(a_{h,l}(t), b_{h,l}\big)\Big]. \tag{30}$$

Comparing this to the continuity equation $\partial_t f_t(x) = -\partial_x(\text{Flux})$, we identify the flux term:

$$f_t(x)v(x, t) = -t_{\max}\,\partial_a I_x\big(a_{h,l}(t), b_{h,l}\big). \tag{31}$$

Substituting $v(x, t) = c_h^{(l)}(x, t)(1 - x)$ and solving for $c_h^{(l)}(x, t)$:

$$c_h^{(l)}(x, t) = -t_{\max}\frac{\partial_a I_x\big(a_{h,l}(t), b_{h,l}\big)}{f_t(x)(1 - x)}. \tag{32}$$

Expanding $f_t(x)$ explicitly yields the form presented in the main text:

$$c_h^{(l)}(x, t) = -t_{\max}\frac{\partial_a I_x\big(a_{h,l}(t), b_{h,l}\big)\,B\big(a_{h,l}(t), b_{h,l}\big)}{x^{a_{h,l}(t)-1}(1 - x)^{b_{h,l}}}. \tag{33}$$

Finally, we check boundary conditions. The flux must vanish at the simplex boundaries ($x = 0$ and $x = 1$) for probability to be conserved. The regularized beta function $I_x(a, b)$ is 0 at $x = 0$ and 1 at $x = 1$ for all $a, b > 0$. Therefore, its derivative with respect to $a$ is:

$$\partial_a I_0(a, b) = 0, \quad \text{and} \quad \partial_a I_1(a, b) = \partial_a(1) = 0. \tag{34}$$

Thus, the flux is zero at $x = 0$ and $x = 1$. It remains to check the other simplex faces $\{x_j = 0\}$ for $j \neq i$. On such a face, the outward normal is aligned with $-\mathbf{e}_j$ and the normal component of the flux is $(p_t^{(h,l)}u_j)\big|_{x_j=0}$. Since $u_j = -c_h^{(l)}(x_i, t)x_j$, we have $(p_t^{(h,l)}u_j)\big|_{x_j=0} = 0$. More precisely, near $x_j = 0$ the Dirichlet density behaves as $p_t^{(h,l)}(\mathbf{x}) = O(x_j^{\alpha_j^{(h,l)}-1})$ with $\alpha_j^{(h,l)} > 0$, hence $x_j p_t^{(h,l)}(\mathbf{x}) \to 0$ and the flux vanishes on that boundary as well. Therefore the net boundary flux on $\partial S_K$ is zero and mass is conserved. $\qquad\square$

## B.3. Rerouting as KL-regularized selection

**Proposition B.3.** *Fix a lineage $h$ and let $p$ denote the baseline distribution on $\mathcal{X}_h$ at time $t_{\text{int}}$ induced by the base flow. Let $\mathcal{K}$ be a Markov kernel on $\mathcal{X}_h$ (mutation/proposal) and define the mutated proposal law $p^{\text{mut}} := p\mathcal{K}$. Let $J : \mathcal{X}_h \to \mathbb{R}$ be measurable and assume $Z_\beta := \mathbb{E}_{\mathbf{X} \sim p^{\text{mut}}} \big[ \exp(\beta J(\mathbf{X})) \big] < \infty$ for some $\beta > 0$. Define*

$$q_\beta(\mathbf{X}) := \frac{1}{Z_\beta} \exp(\beta J(\mathbf{X})) \, p^{\text{mut}}(\mathbf{X}). \tag{35}$$

*Then: For convenience, we restate the KL-regularized selection objective (Eq. (11)):*

$$\max_q \left\{ \mathbb{E}_{\mathbf{X} \sim q} \big[ J(\mathbf{X}) \big] - \tfrac{1}{\beta} \text{KL} \big( q \,\|\, p^{\text{mut}} \big) \right\}.$$

(i) *(Select) $q_\beta$ is the unique maximizer of the KL-regularized objective in Eq. (11) over probability measures $q$ on $\mathcal{X}_h$ with $q \ll p^{\text{mut}}$. In particular, if $\mathcal{K}$ is the identity kernel (no mutation), then $q_\beta$ coincides with the selection tilt in Eq. (8).*

(ii) *(Amplify) Let $\mathbf{X}^{(1)}, \ldots, \mathbf{X}^{(M)} \overset{\text{i.i.d.}}{\sim} p^{\text{mut}}$ and define normalized weights $\bar{w}_m := \exp(\beta J(\mathbf{X}^{(m)})) / \sum_{j=1}^M \exp(\beta J(\mathbf{X}^{(j)}))$. If $\widetilde{\mathbf{X}}^{(1)}, \ldots, \widetilde{\mathbf{X}}^{(M)}$ are obtained by multinomial resampling, i.e. indices $I_1, \ldots, I_M \overset{\text{i.i.d.}}{\sim} \text{Categorical}(\bar{w}_1, \ldots, \bar{w}_M)$ and $\widetilde{\mathbf{X}}^{(m)} := \mathbf{X}^{(I_m)}$, then for any bounded measurable test function $f$,*

$$\frac{1}{M} \sum_{m=1}^M f(\widetilde{\mathbf{X}}^{(m)}) \xrightarrow[M \to \infty]{\text{in probability}} \mathbb{E}_{\mathbf{X} \sim q_\beta} \big[ f(\mathbf{X}) \big]. \tag{36}$$

*Proof.* We prove parts (i) and (ii) in order.

**Part (i): KL-regularized selection yields the exponential tilt.** For brevity, write $p^{\text{mut}} := p_{t_{\text{int}}}^{\text{mut}}$ and $p^{\text{sel}} := p_{t_{\text{int}}}^{\text{sel}}$, and fix $\beta > 0$. Define $p^{\text{sel}}$ by Eq. (35), so that for $p^{\text{mut}}$-almost every $\mathbf{X}$,

$$\log \frac{p^{\text{sel}}(\mathbf{X})}{p^{\text{mut}}(\mathbf{X})} = \beta J(\mathbf{X}) - \log Z_\beta. \tag{37}$$

For any $q \ll p^{\text{mut}}$ with $\text{KL}(q\|p^{\text{mut}}) < \infty$, we expand the KL divergence to $p^{\text{sel}}$:

$$\text{KL}(q\|p^{\text{sel}}) = \int q(\mathbf{X}) \log \frac{q(\mathbf{X})}{p^{\text{sel}}(\mathbf{X})} \, d\mathbf{X} = \int q(\mathbf{X}) \log \frac{q(\mathbf{X})}{p^{\text{mut}}(\mathbf{X})} \, d\mathbf{X} - \int q(\mathbf{X}) \log \frac{p^{\text{sel}}(\mathbf{X})}{p^{\text{mut}}(\mathbf{X})} \, d\mathbf{X}$$
$$= \text{KL}(q\|p^{\text{mut}}) - \beta \, \mathbb{E}_{\mathbf{X} \sim q} \big[ J(\mathbf{X}) \big] + \log Z_\beta, \tag{38}$$

where we used Eq. (37). Rearranging Eq. (38) gives

$$\mathbb{E}_{\mathbf{X} \sim q}[J(\mathbf{X})] - \tfrac{1}{\beta} \text{KL}(q\|p^{\text{mut}}) = \tfrac{1}{\beta} \log Z_\beta - \tfrac{1}{\beta} \text{KL}(q\|p^{\text{sel}}). \tag{39}$$

Since $\text{KL}(q\|p^{\text{sel}}) \geq 0$ with equality iff $q = p^{\text{sel}}$, the objective in Eq. (11) is maximized uniquely at $p^{\text{sel}}$.

**Part (ii): resampling consistency.** Let $\mathbf{X}^{(1)}, \ldots, \mathbf{X}^{(M)} \overset{\text{i.i.d.}}{\sim} p^{\text{mut}}$ and define weights $\bar{w}_m$ as in the statement. Conditional on the sampled particles, multinomial resampling satisfies

$$\mathbb{E}\left[ \frac{1}{M} \sum_{m=1}^M f(\widetilde{\mathbf{X}}^{(m)}) \, \middle| \, \mathbf{X}^{(1:M)} \right] = \sum_{m=1}^M \bar{w}_m f(\mathbf{X}^{(m)}), \tag{40}$$

and

$$\text{Var}\left( \frac{1}{M} \sum_{m=1}^M f(\widetilde{\mathbf{X}}^{(m)}) \, \middle| \, \mathbf{X}^{(1:M)} \right) = \frac{1}{M} \text{Var}_{I \sim \text{Categorical}(\bar{w})} \big( f(\mathbf{X}^{(I)}) \big) \leq \frac{\|f\|_\infty^2}{M}, \tag{41}$$

where $\|f\|_\infty := \sup_{\mathbf{X}} |f(\mathbf{X})| < \infty$.

Next, write the weighted average in Eq. (40) as a self-normalized importance sampling estimator:

$$\sum_{m=1}^{M} \bar{w}_m f(\mathbf{X}^{(m)}) = \frac{\frac{1}{M}\sum_{m=1}^{M} f(\mathbf{X}^{(m)})\, e^{\beta J(\mathbf{X}^{(m)})}}{\frac{1}{M}\sum_{m=1}^{M} e^{\beta J(\mathbf{X}^{(m)})}}. \tag{42}$$

By the law of large numbers and the assumption $Z_\beta < \infty$, the numerator and denominator in Eq. (42) converge in probability to $\mathbb{E}_{p^{\mathrm{mut}}}[f(\mathbf{X})e^{\beta J(\mathbf{X})}]$ and $\mathbb{E}_{p^{\mathrm{mut}}}[e^{\beta J(\mathbf{X})}] = Z_\beta$, respectively, hence the ratio converges in probability to

$$\frac{\mathbb{E}_{p^{\mathrm{mut}}}[f(\mathbf{X})e^{\beta J(\mathbf{X})}]}{Z_\beta} = \mathbb{E}_{p^{\mathrm{sel}}}[f(\mathbf{X})]. \tag{43}$$

Finally, Eq. (41) implies that the empirical average over resampled particles concentrates around its conditional mean at rate $1/\sqrt{M}$. Combining this concentration with the convergence of the conditional mean to Eq. (43) yields Eq. (36). □

## C. Dataset and Preprocessing

We train and evaluate on Pfam-A multiple sequence alignments (MSAs) (Mistry et al., 2021) using the RP35 (representative proteome) filtered alignment set. Pfam families are curated protein-domain families represented by seed alignments and profile HMMs; full alignments are obtained by searching a sequence database with HMMER. Pfam also distributes reduced-redundancy alignments based on representative proteome sets (RP15/RP35/RP55/RP75), which we use to control alignment size while retaining broad sequence diversity.

**Raw source.** The raw input is the Pfam-A RP35 full-alignment flatfile distributed by Pfam in Stockholm format, which contains one aligned block per family. In the version used here, the raw file contains 15,952 Pfam-A families and 12,937,286 aligned sequences (as indicated by the Stockholm annotations).

**Preprocessing workflow.** We use a two-stage preprocessing pipeline to produce a cleaned per-family aligned FASTA corpus. Let $\mathcal{A}$ denote the 20 standard amino acids. First, we convert the Stockholm file into per-family aligned FASTA files, preserving Pfam-provided alignment columns. We map . and – to a gap symbol –, uppercase all letters, and map any non-standard residue to X. Second, for each family we apply: (i) a length filter that discards families with alignment length outside $[20, 2000]$ columns; (ii) a depth cap that uniformly subsamples to at most 5000 sequences per family; (iii) a column filter that drops alignment columns with gap/unknown fraction $> 0.95$ (counting – and X as gap/unknown); and (iv) a minimum-depth filter that discards families with fewer than 100 remaining sequences. The resulting aligned sequences are over the alphabet $\mathcal{A} \cup \{-, \mathtt{X}\}$, where – denotes alignment gaps and X denotes unknown/non-canonical residues. In training and evaluation, we treat both – and X as missing data and mask them out; in generation, gap positions are kept fixed using a per-family gap mask (Sec. 6).

**Processed dataset statistics.** After preprocessing, we retain 8,886 families and 8,942,518 aligned sequences. Family depths range from 100 to 5000 sequences (median 405), and cleaned alignment lengths range from 20 to 1486 columns (median 159).

**Train/test split.** For each family, we perform a deterministic within-family split, holding out 5% of sequences (at least 1 and at most $N_h-1$) for testing and using the remaining sequences for training.

### C.1. ProtBFN pretraining data (UniProtCC)

ProtBFN (Atkinson et al., 2025) is pretrained on a curated UniProtKB corpus rather than Pfam domain MSAs. Specifically, it uses the January 2024 UniProtKB release filtered to high-confidence proteins (Protein Existence PE$\in \{1, 2, 3\}$, excluding hypothetical/unknown) and length $< 512$, yielding 71M sequences ("UniProtCC"). The official release provides model weights but not training code, so we cannot retrain ProtBFN on Pfam or add family conditioning; in Table 1 we therefore report ProtBFN only as an unconditional reference baseline (foldability, self-consistency, within-set diversity, and novelty relative to UniProtCC), and omit family validity.

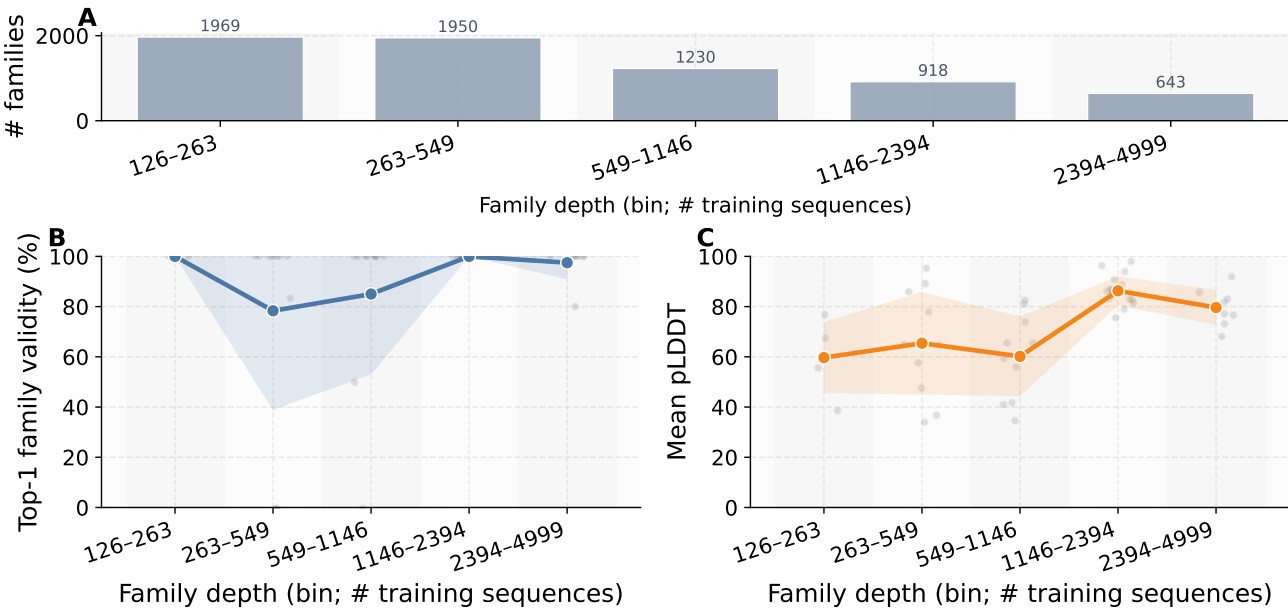

*Figure 6.* **Family depth distribution and performance. (A)** Depth distribution of the processed dataset (number of families per depth bin). **(B–C)** LineageFlow performance versus family depth on the main benchmark: top-1 family validity and mean pLDDT (each dot is a family; shaded bands show mean±std within each bin).

### C.2. PoET pretraining data (UniRef50 homology sets)

PoET (Truong Jr & Bepler, 2023) is pretrained on large-scale sets of homologous sequences derived from UniRef50 rather than Pfam domain MSAs. The PoET paper describes constructing homology sets anchored by UniRef50 representative sequences (e.g., UniRef50 2021_03), retrieving homologs with DIAMOND, filtering out sets with fewer than 10 homologs, and sampling sets with inverse family-size weighting to reduce redundancy. The official PoET release provides inference code and model weights, but does not release training code, the preprocessing pipeline, or the exact training corpus; we therefore evaluate PoET only by sampling from released weights.

## D. Construction of ASR-Based Priors from Pfam MSAs

We construct a family-specific ancestral prior by combining (i) Pfam-provided alignment columns, (ii) maximum-likelihood phylogenetics, and (iii) ancestral sequence reconstruction (ASR) at the root.

**Inputs and preprocessing.** For each family $h$, Pfam provides an MSA of $N_h$ aligned sequences. Let $\mathcal{A}$ denote the $K{=}20$ standard amino acids, and let $A_{n,l}^{(h)}$ be the aligned character at row $n$ and column $l$. We treat gaps and unknown characters as missing data, i.e., $A_{n,l}^{(h)} \in \{\text{-}, \text{X}\}$. We optionally cap the number of sequences used for cleaning (uniform subsampling to at most 5000) and then drop highly missing columns using the per-column missing fraction

$$m_l^{(h)} \;=\; \frac{1}{N_h} \sum_{n=1}^{N_h} \mathbf{1}\Big\{ A_{n,l}^{(h)} \in \{\text{-}, \text{X}\} \Big\},$$

keeping columns with $m_l^{(h)} \leq 0.95$, which defines the alignment length $L_h$. For scalable phylogeny/ASR, we then filter out extremely gappy sequences (gap fraction $> 0.90$) and very short sequences (ungapped length $< 10$), and uniformly subsample at most 400 sequences, yielding a filtered MSA $\widetilde{A}^{(h)} \in (\mathcal{A} \cup \{\text{-}, \text{X}\})^{\widetilde{N}_h \times L_h}$ with the same columns. Here $\widetilde{N}_h$ is the number of retained sequences, and we require $\widetilde{N}_h \geq 4$.

**Tree inference and rooting.** Using $\widetilde{A}^{(h)}$, we infer an unrooted maximum-likelihood phylogenetic tree $\widehat{T}_h^{\text{unroot}}$ with branch lengths under an amino-acid substitution model (in our experiments we use LG (Le & Gascuel, 2008) with discrete gamma rate heterogeneity, LG+$\Gamma_4$) using IQ-TREE (Nguyen et al., 2015). Because standard reversible models identify only

an unrooted tree, we choose a root using Minimal Ancestor Deviation (MAD) rooting (Tria et al., 2017) to obtain a rooted tree $\widehat{T}_h$ with root node $r_h$.

**Root posterior (ASR).** Let $Z_{v,l}^{(h)} \in \mathcal{A}$ be the latent residue at tree node $v$ and alignment column $l$. Fixing $(\widehat{T}_h, \text{model})$, we compute the marginal posterior at the root for each column:

$$p_{\text{root}}^{(h,l)}(a) = \mathbb{P}\left(Z_{r_h,l}^{(h)} = a \,\Big|\, \widetilde{A}^{(h)}, \widehat{T}_h, \text{model}\right), \qquad a \in \mathcal{A}, \; l = 1, \ldots, L_h. \tag{44}$$

This is standard marginal ASR under a continuous-time Markov substitution model (computed via dynamic programming on the tree). In our implementation, we compute these marginal root posteriors with PAML (Yang, 2007) given the rooted tree and fitted substitution model.

**Dirichlet prior parameters.** We map the root posterior to Dirichlet concentration parameters via a "mean–concentration" mapping:

$$\alpha_a^{(h,l)} = \varepsilon + \lambda \, p_{\text{root}}^{(h,l)}(a), \qquad a \in \mathcal{A}. \tag{45}$$

This yields the site-wise prior $q_0(\mathbf{x}^{(l)} \mid h) = \text{Dir}(\mathbf{x}^{(l)}; \boldsymbol{\alpha}^{(h,l)})$ and the product-sequence prior in Eq. (1). In our experiments we use $\lambda = 10$ and $\varepsilon = 10^{-3}$.

Optionally, we mix the ASR prior with a uniform prior of the same total concentration (which shifts the mean toward uniform while keeping the mass fixed):

$$\boldsymbol{\alpha}_{\text{mix}}^{(h,l)} = (1 - \rho) \boldsymbol{\alpha}^{(h,l)} + \rho \frac{\|\boldsymbol{\alpha}^{(h,l)}\|_1}{K} \mathbf{1}, \qquad \rho \sim \text{Unif}[0, \rho_{\max}]. \tag{46}$$

# E. Evaluation Protocols

## E.1. Family Validity via Profile HMMs

In LineageFlow, generation is conditioned on a lineage (family) $h$ and evolves within the corresponding family-specific sequence space $\mathcal{X}_h$. To validate that generated sequences are *biologically consistent* with the intended family—i.e., similar to the training sequences of that family—we evaluate *family membership* using profile hidden Markov models (profile HMMs) (Eddy, 1998) as implemented in HMMER (Eddy, 2011) and distributed by Pfam (Finn et al., 2014).

**Profile-HMM family assignment.** Let $\{\mathcal{M}_h\}_{h \in \mathcal{H}}$ denote the library of family profile HMMs and let $\text{score}(\mathcal{M}_h, \mathbf{s})$ be the HMMER full-sequence bit score for sequence $\mathbf{s}$ under model $\mathcal{M}_h$. For each generated sequence $\mathbf{s}_n$ with intended family label $h_n$ (the family used during generation), we compute its predicted family by scanning against the HMM library:

$$\widehat{h}(\mathbf{s}_n) := \arg\max_{h \in \mathcal{H}} \text{score}(\mathcal{M}_h, \mathbf{s}_n). \tag{47}$$

**Metrics.** We report the top-1 family assignment accuracy

$$\text{Acc}_{\text{fam}} = \frac{1}{N} \sum_{n=1}^{N} \mathbf{1}\left\{\widehat{h}(\mathbf{s}_n) = h_n\right\}, \tag{48}$$

and a hit-rate that counts any significant HMM match at threshold $\tau$ (Hit_any), based on HMMER E-values:

$$\text{Hit}_{\text{any}}(\tau) = \frac{1}{N} \sum_{n=1}^{N} \mathbf{1}\left\{\min_{h \in \mathcal{H}} \text{Evalue}(\mathcal{M}_h, \mathbf{s}_n) \leq \tau\right\}. \tag{49}$$

**Interpretation.** Higher $\text{Acc}_{\text{fam}}$ ($\uparrow$) indicates that generated sequences are most strongly recognized as belonging to the intended family rather than a related family. Higher Hit_any ($\uparrow$) indicates that a larger fraction of sequences achieve at least one statistically significant HMM match (at threshold $\tau$), independent of the intended family.

### E.2. Novelty via Nearest-Neighbor Similarity

To assess whether generated sequences are genuinely new (rather than near-copies of training examples), we measure *nearest-neighbor sequence similarity* to the training corpus using MMseqs2 (Steinegger & Söding, 2017), following common practice in protein generative modeling (Alamdari et al., 2023). Because "novelty" can be trivially increased by generating low-quality sequences, we report novelty *conditioned on foldability*: all novelty metrics are computed on the subset of sequences with mean pLDDT$\geq 70$ (Appendix E.3). Let $\mathcal{R}_{\text{all}} := \bigcup_{h \in \mathcal{H}} \mathcal{R}_h$ denote the *global* reference set consisting of all training sequences across families (after the same preprocessing and split used for training).

**Nearest-neighbor identity.** For each (foldable) generated amino-acid sequence $\mathbf{s}_n$, we find its best-matching reference sequence in $\mathcal{R}_{\text{all}}$ and extract the reported pairwise sequence identity $\text{Id}(\mathbf{s}, \mathbf{r}) \in [0, 1]$ from the alignment.[1] We define the global nearest-neighbor identity as

$$\text{NNId}(\mathbf{s}_n) := \max_{\mathbf{r} \in \mathcal{R}_{\text{all}}} \text{Id}(\mathbf{s}_n, \mathbf{r}). \tag{50}$$

**Summary metrics and interpretation.** We report the mean±std of NNId and thresholded novelty rates, computed over sequences for which MMseqs2 returns at least one hit passing the coverage thresholds,

$$\text{Novelty@}\delta := \frac{1}{N} \sum_{n=1}^{N} \mathbf{1}\{\text{NNId}(\mathbf{s}_n) < \delta\}, \tag{51}$$

for $\delta \in \{0.80, 0.60\}$, computed over the foldable subset (pLDDT$\geq 70$); here $N$ denotes the number of foldable sequences with at least one MMseqs2 hit. If a method yields no MMseqs2 hits in the foldable subset, NNId and Novelty@$\delta$ are undefined; we mark this case as — in Table 1. Lower NNId ($\downarrow$) and higher Novelty@$\delta$ ($\uparrow$) indicate greater novelty relative to training data; these should be interpreted together with family-validity (Appendix E.1) to ensure samples remain within the intended family manifold.

**Diversity via clustering.** To summarize within-set diversity, we cluster the foldable subset (pLDDT$\geq 70$) using MMseqs2 `cluster` at 80% sequence identity with minimum coverage 0.8 on ungapped amino-acid sequences. We report Diversity@0.8 as the number of clusters; higher values indicate more diverse generations.

### E.3. Foldability and Self-Consistency

Following EvoDiff-style evaluation (Alamdari et al., 2023), we report two complementary metrics: (i) *foldability* via predicted structure confidence, and (ii) *self-consistency* via inverse folding likelihood under the predicted backbone.

**Foldability via OmegaFold pLDDT.** For each generated amino-acid sequence $\mathbf{s}_n$ of length $L_n$, we predict a backbone structure $\widehat{\mathbf{Y}}_n$ using OmegaFold (Wu et al., 2022). OmegaFold outputs a per-residue confidence score (pLDDT) $\text{pLDDT}_{n,j} \in [0, 100]$ for residue $j$. We summarize foldability by the mean pLDDT,

$$\text{Fold}(\mathbf{s}_n) := \frac{1}{L_n} \sum_{j=1}^{L_n} \text{pLDDT}_{n,j}. \tag{52}$$

Higher values ($\uparrow$) indicate more confident predicted structures and are commonly used as a proxy for foldability.

**Self-consistency via inverse folding perplexity.** Given the predicted backbone $\widehat{\mathbf{Y}}_n$, we score the original sequence $\mathbf{s}_n$ under an inverse folding model $p_\phi(\mathbf{s} \mid \widehat{\mathbf{Y}})$ (we use ESM-IF (Hsu et al., 2022)). Let $p_\phi(s_{n,j} \mid \widehat{\mathbf{Y}}_n, s_{n,<j})$ be the autoregressive conditional probability assigned to residue $s_{n,j}$ at position $j$. We compute the per-residue negative log-likelihood and the corresponding self-consistency perplexity:

$$\text{NLL}_{\text{sc}}(\mathbf{s}_n) := -\frac{1}{L_n} \sum_{j=1}^{L_n} \log p_\phi\left(s_{n,j} \mid \widehat{\mathbf{Y}}_n, s_{n,<j}\right), \qquad \text{scPPL}(\mathbf{s}_n) := \exp\left(\text{NLL}_{\text{sc}}(\mathbf{s}_n)\right). \tag{53}$$

---

[1] We compute novelty on ungapped amino-acid sequences and require a minimum alignment coverage for both query and target (e.g., 80%) to avoid short high-identity local matches.

Lower values ($\downarrow$) indicate that the predicted backbone is more compatible with the original sequence under a structure-conditioned sequence model, closing the loop "fold $\rightarrow$ score under inverse folding."

