# OpenReview forum: "LineageFlow: Flow Matching for High-Fidelity Family-Aware Protein Sequence Generation"
_ICML.cc/2026/Conference — ICML 2026 regular_

### Official Review · Reviewer_84Mh · 2026-03-04

**Soundness:** 3
**Presentation:** 3
**Significance:** 3
**Originality:** 3
**Overall Recommendation:** 4
**Confidence:** 3

**Summary:**

The authors propose a novel approach called LineageFlow, which enables the generation of family-specific protein sequences by starting from a lineage prior instead of random or mask tokens. They also propose a rerouting strategy that guides the sampling towards high plausibility.

**Compliance With Llm Reviewing Policy:**

Affirmed.

**Final Justification:**

I think this is in general a good paper with novel contribution and good presentation. The authors have addressed my concerns raised in the review during their rebuttal. However, I am not familiar with the importance of family-aware protein sequence generation.

**Key Questions For Authors:**

1. From Table 1, it seems the proposed rerouting improve the family validity, foldability, diversity and ppl but hurt the novelty. Could the author explain why?
  2. Could the lineageFlow generalize to unseen protein family not in the training set?
  3. Could you add more recent sequence generation baselines (such as ESM3[1], DPLM[2], DPLM2[3], etc) to comparison to contextualize the LineageFlow?
  4. The inference speed in Table 2 of LineageFlow looks not very attractive. Could the authors explain what cause the proposed model being slower than most of the baselines except for ProtBFN[4]? Any idea to improve this?

[1] Hayes, Thomas, Roshan Rao, Halil Akin, Nicholas J. Sofroniew, Deniz Oktay, Zeming Lin, Robert Verkuil et al. "Simulating 500 million years of evolution with a language model." Science 387, no. 6736 (2025): 850-858.

[2] Wang, Xinyou, Zaixiang Zheng, Fei Ye, Dongyu Xue, Shujian Huang, and Quanquan Gu. "Diffusion language models are versatile protein learners." arXiv preprint arXiv:2402.18567 (2024).

[3] Wang, Xinyou, Zaixiang Zheng, Fei Ye, Dongyu Xue, Shujian Huang, and Quanquan Gu. "Dplm-2: A multimodal diffusion protein language model." arXiv preprint arXiv:2410.13782 (2024).

[4] Atkinson, Timothy, Thomas D. Barrett, Scott Cameron, Bora Guloglu, Matthew Greenig, Charlie B. Tan, Louis Robinson, Alex Graves, Liviu Copoiu, and Alexandre Laterre. "Protein sequence modelling with Bayesian flow networks." Nature Communications 16, no. 1 (2025): 3197.

**Limitations:**

Yes

**Strengths And Weaknesses:**

Pros
  1. This is an innovative application of a specialized flow matching on protein sequences with good motivation.

Cons
  1. The training of the model requires family-specific PSSMs, which is a strong prior and may introduce additional biases compared to purely data-driven sequence language models, which may limit the utility of this method.
  2. There are missing baselines for the sequence generation benchmarks. For example, ESM3 and DPLM are competitive baselines for sequence generation.

---

> ### Author Rebuttal · Authors · 2026-03-29
>
> We thank the reviewer for the thoughtful feedback and address the comments below.
>
> ---
>
> ### **Comment 1**
> > Why does rerouting improve validity/foldability/diversity/ppl but reduce novelty?
>
> ### **Response.**
> 1. **Main point.** This is an expected trade-off: rerouting is designed to improve **plausibility**, so a **moderate drop in novelty** is natural. We will clarify this trade-off in the revision.
>
> 2. **Why this happens.** Rerouting optimizes a soft-masked **ESM2 plausibility score** $J(\mathbf{X})$ via **KL-regularized selection** (Eq. 9). Since ESM2 is trained on natural proteins, higher $J(\mathbf{X})$ favors sequences closer to the natural family manifold, improving family validity, foldability, and ESM-IF self-consistency, while reducing distance to the training corpus.
>
> 3. **Why diversity is preserved.** Rerouting is **not pure** score maximization: the **KL term** keeps the selected distribution close to the base-flow proposal, mitigating novelty loss and preventing collapse.
>
> ### **Comment 2**
> > Can LineageFlow generalize to unseen protein families?
>
> ### **Response.**
> Yes.
>
> 1. In `Sec. 6.2`, we **hold out three enzyme families from denoiser training** and generate for them **without family-specific fine-tuning**. At inference, LineageFlow still uses the corresponding family-specific ASR prior.
>
> 2. Even in this setting, generated sequences recover family-specific properties close to real sequences, including **motif agreement, solubility, and thermostability**, while maintaining reasonable novelty.
>
> These results support generalization beyond families seen during denoiser training, provided an inference-time family prior is available.
>
> ### **Comment 3**
> > Please compare with recent generators such as ESM3, DPLM, and DPLM2.
>
> ### **Response.**
> Following your recommendation, we added three comparisons:
>
> 1. **Official checkpoints.** We sampled from the **official ESM3-open** and **official DPLM** checkpoints and evaluated them with our pipeline.
>
> 2. **Controlled baseline.** We trained a **family-conditioned DPLM** on our Pfam-based corpus.
>
> 3. **Scaling study.** We trained LineageFlow on a **larger Pfam-based corpus**.
>
> Because **official ESM3/DPLM** use different corpora, we present them as **contextual references**, not fully matched baselines.
>
> | Method | Training corpus | Family conditioning | Top-1 family acc. | pLDDT | scPerplexity | Novelty@0.8 | Novelty@0.6 |
> | --- | --- | --- | ---: | ---: | ---: | ---: | ---: |
> | ESM3-open | unknown | none | - | 65.51 | 7.96 | -| - |
> | DPLM | ~41.5M seqs | none | 0.0 | 80.69 | 5.36 | 97.9 | 67.3 |
> | DPLM | ~7.8M seqs| family label | 0.1|  73.28|  7.58|  96.3|  55.5|
> | LineageFlow | ~7.8M seqs | ASR prior | 95.3 |  76.6|  6.67|  89.6|  52.0|
> | LineageFlow| ~23.6M seqs | ASR prior | 95.1 |  79.1|  5.65| 97.3 | 60.2 |
>
> Overall, recent large generators can produce protein-like sequences, but **family-specific control** remains the key challenge, which is precisely the regime LineageFlow targets. We will include these results in the revision.
>
> ### **Comment 4**
> > Why is LineageFlow slower than most baselines, and how can it be accelerated?
>
> ### **Response.**
> Table 2 reflects a **quality-speed trade-off**, not a claim of fastest inference.
>
> 1. **Why slower than DFM.** Even without rerouting, LineageFlow is slower than DFM (`759.05s` vs. `573.63s` for 512 sequences) because DFM uses a simple family-agnostic `Dir(1)` prior, whereas LineageFlow uses a **family-specific ASR prior** and **family-specific vector field**, making initialization and vector field computation more expensive.
>
> 2. **Extra rerouting cost.** Rerouting adds `287.78s` (`1046.83s` vs. `759.05s` for 512 sequences) through extra mutate-select-amplify operations.
>
> 3. **Possible acceleration.** Clear directions include **faster `betainc` approximation** (in vector field computation), **fewer ODE steps**, and **smaller rerouting populations/fewer rounds**.
>
> We will clarify this discussion in the revision.
>
> ### **Comment 5**
> > Family-specific PSSMs are strong and may limit the utility of this method.
>
> ### **Response.**
> We agree this is a limitation and discussed it in the Conclusion and Limitations section.
>
> 1. LineageFlow is **not** intended for fully unconditional protein generation; it targets **family-aware generation**.
>
> 2. We believe this regime remains broad and practically important, since many natural proteins have homologs and family context. For example, **Pfam 38.1** contains **27,481 families** and **69.6M sequences**.
>
> 3. Within this setting, we compare against baselines with conditioning of different strengths, from **family labels** to the stronger **family-MSA prompt** used by PoET. Despite these signals, **LineageFlow is the only method that achieves top-1 family accuracy close to natural sequences**, which is the central objective of the paper.
> ---
> We thank the reviewer again for the helpful questions and suggestions. We welcome any further questions if additional clarification needed!

---

> > ### Author Rebuttal · Reviewer_84Mh · 2026-04-04
> >
> > Thank the authors for their response. I will increase my score accordingly by increasing the significance to 3.

---

> > > ### Author Response · Authors · 2026-04-04
> > >
> > > Dear Reviewer 84Mh,
> > >
> > > We are very glad to hear that your concerns have been adequately addressed, and we sincerely appreciate your decision to increase the significance score.
> > >
> > > Your constructive feedback has been very helpful in improving the paper, and we will make sure to incorporate your comments into the revision.
> > >
> > > Thank you again for your time and consideration. Wishing you a good day ^_^
> > >
> > > Best regards,
> > >
> > > The Authors

---

### Official Review · Reviewer_FMCV · 2026-03-11

**Soundness:** 2
**Presentation:** 3
**Significance:** 3
**Originality:** 3
**Overall Recommendation:** 4
**Confidence:** 2

**Summary:**

This paper proposes a Dirichlet flow-matching model defined on the probability simplex. It discards traditional uniform noise or mask initialization in favor of phylogenetic priors derived from Ancestral Sequence Reconstruction (ASR). It also introduces a training-free _rerouting_ mechanism for target guidance.

**Compliance With Llm Reviewing Policy:**

Affirmed.

**Key Questions For Authors:**

- For highly engineered scaffolds or targets with distinct evolutionary trajectories, natural MSAs are often either too broad or heavily biased. How robust is the ASR prior when the target falls into a sparsely populated or highly skewed region of the phylogenetic tree?

- The rerouting mechanism is elegant, but does the mutation-selection-amplification step introduce any bias toward local sequence minima, potentially reducing the global diversity of the generated family?

**Strengths And Weaknesses:**

Strength

- Framing sequence generation as _structured mutation from an evolved scaffold_ rather than _denoising from scratch_ drastically improves the fidelity of family-conditioned generation.

- Introducing a "mutation-selection-amplification" (akin to directed evolution) mechanism at intermediate timesteps elegantly achieves guided sampling without requiring classifier gradients at every step.

Weakness

- Heavy reliance on MSA quality and the computational overhead of ASR limits the framework's utility for de novo design of entirely novel folds or orphan proteins.

- The approach is strictly family-aware, meaning it cannot easily traverse the fitness landscape between distinct protein families to invent new multi-domain architectures.

---

> ### Author Rebuttal · Authors · 2026-03-29
>
> We thank the reviewer for the careful reading and constructive suggestions. Below we respond to the main points.
>
> ---
> ### **Comment 1**
> > Heavy reliance on MSA quality and the computational overhead of ASR limits the framework's utility for de novo design of entirely novel folds or orphan proteins.
>
> ### **Response.**
> The MSA requirement is a limitation, and we explicitly acknowledge it in the Conclusion and Limitations section. However, our method is generally robust to MSA quality and depth, as shown table below. We also clarify the scope and ASR overhead.
>
> 1. LineageFlow is **not** intended for fully unconditional generation like generic protein language models; it is designed for **family-aware generation**. We believe this regime remains broad and important, since many natural proteins have homologs and family context (e.g., Pfam 38.1 contains **27,481 families** and **69.6M sequences**).
>
> 2. The ASR step is a **one-time family-level preprocessing cost**, not a per-sample cost, so it can be amortized across many generations for the same family.
>
> 3. We also provide evidence beyond training families in **Sec. 6.2**, where we hold out enzyme families during denoiser training and still generate meaningful family-consistent sequences from their inference-time priors **without family-specific fine-tuning**.
>
> We will clarify in the revision that LineageFlow is intended for this family-aware regime, rather than as a universal method for orphan proteins or entirely novel folds.
>
> ### **Comment 2**
> > The approach is strictly family-aware, meaning it cannot easily traverse the fitness landscape between distinct protein families to invent new multi-domain architectures.
>
> ### **Response.**
> 1. LineageFlow does **not** train a separate model for each family. It uses **one shared denoiser trained jointly across families**, while family specificity enters only through the **inference-time lineage prior**. This is also reflected in **Sec. 6.2**, where the same denoiser generalizes to **held-out families** without family-specific fine-tuning.
>
> 2. That said, the current paper focuses on **high-fidelity generation within known target families**, not invention of new multi-domain architectures. We agree this would be an interesting future direction.
>
> ### **Question 1**
> > How robust is the ASR prior when the target falls into a sparsely populated or highly skewed region of the phylogenetic tree?
>
> ### **Response.**
> We agree that robustness under sparse or biased homolog data is an important concern.
>
> 1. LineageFlow is not simply replaying the prior: the prior-only ASR/PSSM baseline is substantially weaker than LineageFlow, and training already exposes the model to imperfect priors through **prior weakening and randomization**. For efficiency, our ASR prior construction also uses only a **moderate subsample** of each family MSA.
>
> 2. To address this more directly, we conducted a focused ablation with the **trained model frozen** and only changed prior construction on representative families. We considered two stress settings: **(i) sparse priors**, rebuilt from only `50` homologs, and **(ii) skewed priors**, rebuilt from a deliberately biased MSA subset (nearest neighbors of a random seed sequence). We then ran the same generation and evaluation pipeline as in the main paper and compared against the original prior.
>
> | Prior setting | Top-1 family acc. | pLDDT | scPpl | Diver.|
> | --- | ---: | ---: | ---: | ---: |
> | Original prior (`200` homologs) | 95.3|76.6  | 6.67 |587|
> | Sparse prior (`50` homologs) | 88.5 | 73.5 | 7.68 | 502|
> | Skewed prior  (`200` homologs)  | 93.2 | 75.1 | 7.34 | 475|
>
> Overall, the results suggest **robustness to moderately sparse and skewed priors**.
>
> ### **Question 2**
> > The rerouting mechanism is elegant, but does the mutation-selection-amplification step introduce any bias toward local sequence minima, potentially reducing the global diversity of the generated family?
>
> ### **Response.**
> Our paper already provides both theoretical and empirical evidence that rerouting does not simply collapse samples into local minima.
>
> 1. **Theoretically**, rerouting is **not a greedy search** for a single best sequence. It is a mutate-select-amplify operator with a **KL-regularized** selection objective (Eq. 9): **mutation explicitly injects diversity**, and selection is regularized relative to the mutated proposal. Thus, it biases the population toward higher-fitness regions without collapsing to a single mode.
>
> 2. **Empirically**, Table 1 shows that, relative to LineageFlow **without rerouting**, rerouting improves **top-1 family accuracy** (`93.0 -> 95.3`), **pLDDT** (`69.6 -> 76.6`), and **scPerplexity** (`7.96 -> 6.67`), while **diversity also increases** (`440 -> 587` clusters).
>
> Overall, these results suggest that rerouting improves plausibility and family fidelity without causing mode collapse.
>
> ---
>
> We thank the reviewer again for the thoughtful feedback. We are happy to provide further clarification if needed!

---

### Official Review · Reviewer_eYYd · 2026-03-13

**Soundness:** 3
**Presentation:** 3
**Significance:** 3
**Originality:** 3
**Overall Recommendation:** 5
**Confidence:** 4

**Summary:**

The paper proposes flowmatching based a protein sequence generation metho. It incorporates evolutionary information into discrete flow matching by initializing generation from ancestral sequence reconstruction-based priors rather than uniform noise. The model learns to generate sequences within protein families using dirichlet flow matching, and introduces a rerouting procedure at inference time that applies a mutate-select-amplify step to guide generation toward desired objectives. Experiments on large Pfam datasets show that the approach generates diverse sequences that remain consistent with the target protein family while improving structural plausibility and novelty compared to several baseline models.

**Compliance With Llm Reviewing Policy:**

Affirmed.

**Final Justification:**

My concerns are addressed. Increasing the score to accept.

**Key Questions For Authors:**

Please see the weaknesses above.

**Limitations:**

Yes.

**Strengths And Weaknesses:**

Strengths:

1. The authors emphasize of leveraging explicit biological prior (in the form of ASR), which is interesting and a potentially helpful direction.
2. The usage of "rerouting", which is a form of filtering less scoring samples and amplifying the higher scoring ones, is often an effective one for iterative (e.g., diffusion and flow matching based) approaches.
3. They conducted evaluation on a large-scale benchmark.
4. experimental results show improvement in a wide range of metrics and settings.

Weakness:
1. My main concern is the limited methodological novelty. The paper uses flow matching based seq generation with guidance from external/additional tools, with the prior from ASR. I would suggest the authors clarify on their actual methodic novelty.
2. While the authors conducted a good set of experiments and evaluations, the metrics they emphasized come from predictive models (the foldability metrics related for examples) or are heuristics based.
3. It is not clear if the improvements come from the proposed method or more enriched datasets used for optimization.

---

> ### Author Rebuttal · Authors · 2026-03-29
>
> We thank the reviewer for the careful reading of our submission and the constructive comments. Below we provide point-by-point responses.
>
> ---
>
> ### **Comment 1**
> > My main concern is the limited methodological novelty. The paper uses flow matching based seq generation with guidance from external/additional tools, with the prior from ASR. I would suggest the authors clarify on their actual methodic novelty.
>
> ### **Response.**
> We thank the reviewer for this important comment. We agree that the methodological novelty should be stated more clearly. We do **not** claim novelty in flow matching, ASR, or external fitness objectives by themselves. Rather, the novelty lies in integrating them into a **new family-aware generative formulation**.
>
> 1. **Family conditioning through the prior, not the denoiser.** Prior work typically achieves family-aware generation by conditioning the denoiser directly, e.g., learning $f(x_t,h)$, where $h$ is a family label or stronger context such as an MSA prompt. In LineageFlow, family specificity instead enters through the **lineage prior** $q_0^{(h)}$, while the denoiser is **shared across all families**. This differs from standard family-label or MSA-prompt conditioning. Empirically, this matters: LineageFlow reaches family validity close to natural sequences ($95.3$ top-1 family accuracy vs. $96.6$ for held-out natural sequences), whereas the family-label and MSA-prompt baselines in Table 1 both obtain $0.0$ top-1 family accuracy.
>
> 2. **Rerouting as a single intermediate-time inference operator.** To improve sample quality, guided diffusion / flow-matching methods typically either guide sampling at **every step**, e.g., $\hat v(x_t,t) + \lambda \nabla_x J(x_t)$. In contrast, our rerouting is a **single intermediate-time mutate-select-amplify intervention**. Inspired by directed evolution and tailored to the family-aware protein setting, this single-time intervention is simple, low-cost, and effective.
>
> > **Remark:** These components are also empirically isolated in the paper. **DFM** uses the same denoiser backbone and flow-matching framework as LineageFlow, so the large performance gap highlights the contribution of the lineage prior and rerouting.
>
> We also added an ablation comparing single-step rerouting with per-step guidance (**using the same ESM2 scorer**).
>
> | Method | Fam. acc. | pLDDT | scPPL | Nov.@0.8 | Div. | Time (512 seqs) |
> | --- | ---: | ---: | ---: | ---: | ---: | ---: |
> | Single-step rerouting | 95.3 | 76.6 | 6.67 | 86.2 | 587 | 1046s |
> | Per-step guidance | 91.2 | 78.6 | 6.14 | 89.7 | 468 | 6903s |
>
> Overall, per-step guidance is about **6.6×** slower, slightly worse on family fidelity and diversity, and better on foldability and self-consistency. This suggests our **single-step rerouting is an effective and efficient alternative to standard per-step guidance**.
>
> ### **Comment 2**
> > Metrics come from predictive models (the foldability metrics related for examples) or are heuristics based.
>
> ### **Response.**
> We thank the reviewer for this important comment. We fully agree that in silico metrics cannot substitute for experimental validation, and **we explicitly acknowledge this limitation in the Conclusion and Limitations section.**
>
> At the same time, because wet-lab validation is not available in this work, we aimed to make the evaluation as rigorous as possible within this practical constraint, which is **standard practice in protein generation papers**. Specifically, we evaluate generated sequences from multiple complementary perspectives:
>
> - Foldability-related metrics (pLDDT and self-consistency), to assess structural plausibility;
> - Family validity, using profile-HMM recognition;
> - Novelty, using mmseqs search against the training corpus;
> - Diversity, using distributional statistics over generated samples.
>
> Therefore, LineageFlow improves consistently across multiple evaluation axes, which together provide stronger evidence that the observed gains are substantive.
>
> ### **Comment 3**
> > It is not clear if the improvements come from the proposed method or more enriched datasets used for optimization.
>
> ### **Response.**
> - We respectfully note that **Table 1 already provides a controlled comparison** showing that the gains are *not* due to a more enriched optimization dataset. In particular, **LineageFlow, DFM, and EvoDiff are trained on the same Pfam training corpus**.
> - Among these baselines, DFM is the most direct control / ablation of LineageFlow: it uses the same denoiser architecture and Dirichlet flow-matching framework, but removes the two proposed ingredients, namely the **lineage-informed ASR prior** and the **intermediate rerouting** step. Therefore, DFM’s weaker performance supports that the **improvement comes from the proposed method itself**, rather than from differences in training data.
>
> ---
> We sincerely thank the reviewer again for the careful reading and constructive feedback. We would be glad to clarify any further details if needed!

---

> > ### Author Rebuttal · Reviewer_eYYd · 2026-04-03
> >
> > My concerns are resolved. Increasing score to accept.

---

> > > ### Author Response · Authors · 2026-04-04
> > >
> > > Dear Reviewer eYYd,
> > >
> > > We are very glad to hear that our responses have addressed your concerns, and we sincerely appreciate your updated score. Your constructive feedback has been very helpful in improving the paper. We will make sure to incorporate your comments and suggestions into the revision.
> > >
> > > Thank you again for your time and thoughtful evaluation.
> > >
> > > Wishing you a good day!
> > >
> > > Best regards,
> > >
> > > The Authors

---

### Official Review · Reviewer_6Bek · 2026-03-13

**Soundness:** 3
**Presentation:** 3
**Significance:** 3
**Originality:** 4
**Overall Recommendation:** 4
**Confidence:** 3

**Summary:**

This paper introduces LineageFlow, a Dirichlet flow-matching model. It initializes generation from lineage priors encoded from ancestral sequence reconstruction. The generation process is a structured mutation from an evolved scaffold. The single-step rerouting steers samples toward objectives while empirically preserving family validity. Experimental results demonstrate the efficiency of the proposed method.

**Compliance With Llm Reviewing Policy:**

Affirmed.

**Final Justification:**

My main concerns about the lineage prior and the larger-scale pretraining ablation are resolved, so I keep my positive score of 4.

**Key Questions For Authors:**

See weakness

**Limitations:**

yes

**Strengths And Weaknesses:**

Strength:

1. The method is novel and well-motivated. For proteins, many sites are highly conserved to maintain structural integrity and biochemical function, so replacing the generic priors with the ancestral priors is reasonable.

2. The experimental settings are fairly comprehensive, and the analysis study is promising.



Weakness:

1. As mentioned in the strength, the lineage prior is the core of the method. A complete sensitivity analysis for its construction choices is needed.

2. “Zero-shot” of enzyme generation is somehow overclaimed. Although the authors froze the denoiser on the held-out enzyme families, LineageFlow still constructs family priors from the target family MSA and phylogenetic tree during inference.

3. As described in Table 1, the comparison to ProtBFN is informative but not controlled (ProtBFN uses an 8x larger corpus). If feasible, the authors could add a larger-scale pretraining ablation study for the LineageFlow backbone.

---

> ### Author Rebuttal · Authors · 2026-03-29
>
> We sincerely thank the reviewer for the careful reading of our submission and the constructive feedback. Below we provide our responses.
>
> ---
>
> ### **Comment 1**
> > As mentioned in the strength, the lineage prior is the core of the method. A complete sensitivity analysis for its construction choices is needed.
>
> ### **Response.**
> We thank the reviewer for this insightful comment. In earlier development, we examined this issue at two levels, and we now clarify both the rationale for adopting an ASR-based prior and the implementation choices used in practice.
>
> - **First, at the method level**, we evaluated simpler family priors based on **raw MSA frequencies and PSSM-style statistics**. While these variants could still generate sequences with reasonable foldability (e.g., pLDDT), they consistently yielded **lower novelty and diversity than the ASR-based prior**. In particular, their novelty was typically **10–15 percentage points below LineageFlow**, whose ASR prior achieves **89.6 / 52.0** for `Novelty@0.8 / 0.6`. A likely reason is that such priors remain closer to the empirical training distribution and therefore provide a less informative evolutionary starting point.
>
> - **Second, at the implementation level**, our ASR prior is constructed using a **standard, practical pipeline**: starting from the family MSA, we subsample to a moderate size, then infer the phylogeny and marginal ancestral posteriors with **IQ-TREE** under a standard maximum-likelihood setup (LG+G4, `-fast` in our implementation). We did not exhaustively tune ASR software or hyperparameters. That strong results are obtained with this efficient, off-the-shelf pipeline suggests that the method does not rely on unusually delicate implementation choices.
>
> Due to the rebuttal-time constraint, we could not run a broader re-construction ablation over multiple ASR settings, especially since these tools are largely CPU-based. Still, the fact that **moderate MSA subsampling** and **practical fast ASR tools** already yield strong performance gives us confidence that the approach is reasonably robust to prior construction choices. We will clarify this in the revision.
>
> ### **Comment 2**
> > Zero-shot” of enzyme generation is somehow overclaimed. Although the authors froze the denoiser on the held-out enzyme families, LineageFlow still constructs family priors from the target family MSA and phylogenetic tree during inference.
>
> ### **Response.**
> We agree that precise wording is important in scientific writing, and we thank the reviewer for pointing this out.
>
> 1. Our intent was **not** to claim that the held-out enzyme families are entirely information-free at inference. Rather, the intended claim is that the **learned denoiser never sees these families during training**, and **no family-specific fine-tuning** is performed.
>
> 2. We therefore agree that the phrase **“zero-shot enzyme generation”** is imprecise and potentially misleading. In the revision, we will replace it with a more accurate description, such as **“held-out-family generation without fine-tuning.”**
>
> ### **Comment 3**
> > As described in Table 1, the comparison to ProtBFN is informative but not controlled (ProtBFN uses an 8x larger corpus). If feasible, the authors could add a larger-scale pretraining ablation study for the LineageFlow backbone.
>
> ### **Response.**
> Thank you for this valuable suggestion. Following it, **we trained LineageFlow on a larger Pfam dataset** (**~23.6M** sequences vs. **~7.8M** in the paper). The resulting model shows improved generation quality, especially in **foldability**, **novelty**, and **diversity**, providing additional evidence that LineageFlow continues to benefit from richer training data.
>
> | Method | Training corpus | Top-1 family acc. | pLDDT | scPerplexity | Novelty@0.8 | Novelty@0.6 | Diversity |
> | --- | --- | ---: | ---: | ---: | ---: | ---: | ---: |
> | LineageFlow | ~7.8M seqs | 95.3 | 76.6 | 6.67 | 89.6 | 52.0 | 587 |
> | LineageFlow | ~23.6M seqs | 95.1 | 79.1 | 5.65 | 97.3 | 60.2 | 643 |
>
> We will include these additional results and discussion in the revision. We thank the reviewer again for this helpful suggestion.
>
> ----
>
> We thank the reviewer again for the thoughtful comments! We hope our responses have addressed the main concerns. We would be happy to further clarify any remaining points if needed!

---

> > ### Author Rebuttal · Reviewer_6Bek · 2026-04-02
> >
> > Thanks to the author's effort and thoughtful responses. Most of my concerns are resolved. I choose to keep my positive score.

---

> > > ### Author Response · Authors · 2026-04-03
> > >
> > > Dear Reviewer 6Bek,
> > >
> > > We greatly appreciate your constructive comments and are pleased that our rebuttal resolved your concerns!
> > >
> > > Thank you for your time and for maintaining your positive score. We will be sure to integrate your valuable suggestions into the final version of the manuscript.
> > >
> > > Best regards,
> > >
> > > The Authors

---

### Decision · Program_Chairs · 2026-04-30

**Decision:**

Accept (regular)

**Comment:**

LineageFlow introduces a novel and effective approach to family-aware protein sequence generation by replacing standard uniform noise priors with biologically grounded ancestral sequence reconstruction. During the rebuttal phase, the authors successfully resolved initial concerns regarding the method's computational overhead and generalization capabilities. Given its strong empirical performance across foldability, novelty, and structural plausibility metrics, this paper offers a significant contribution.